# Increasing associative plasticity in temporo-occipital back-projections improves visual perception of emotions

Sara Borgomaneri [1] ✉, Marco Zanon [1,2], Paolo Di Luzio [1], Antonio Cataneo [1], Giorgio Arcara [3], Vincenzo Romei [1,4], Marco Tamietto [5,6] ✉ & Alessio Avenanti [1,7] ✉

The posterior superior temporal sulcus (pSTS) is a critical node in a network specialized for perceiving emotional facial expressions that is reciprocally connected with early visual cortices (V1/V2). Current models of perceptual decision-making increasingly assign relevance to recursive processing for visual recognition. However, it is unknown whether inducing plasticity into reentrant connections from pSTS to V1/V2 impacts emotion perception. Using a combination of electrophysiological and neurostimulation methods, we demonstrate that strengthening the connectivity from pSTS to V1/V2 selectively increases the ability to perceive facial expressions associated with emotions. This behavior is associated with increased electrophysiological activity in both these brain regions, particularly in V1/V2, and depends on specific temporal parameters of stimulation that follow Hebbian principles. Therefore, we provide evidence that pSTS-to-V1/V2 back-projections are instrumental to perception of emotion from facial stimuli and functionally malleable via manipulation of associative plasticity.

Humans excel in perception of emotion from other people's facial expressions, an ability fundamental for effective social interactions and linked to situations ancestrally relevant for survival[1]. Traditionally, neuroscience research has documented enhanced responses to facial expressions in visual areas[2,3]. However, this activity has often been interpreted as consequential, rather than integral, to emotional appraisal, which has been assumed to occur elsewhere in the brain (e.g., in the limbic system)[4,5]. Yet, recent evidence indicates that sensory representations, measured as patterns of activity confined within the visual system, can be sufficient to accurate perception of emotions[6–8]. Functional and connectional properties converge toward a specialized and partly segregated pathway for facial expression

recognition that begins in the early visual cortex (V1/V2), and terminates downstream in the posterior banks of the superior temporal sulcus (pSTS)[9,10]. For example, activity in V1/V2 and pSTS selectively responds to reward signals, predicts category-specific emotion perception, and encodes various affective states according to several gradients[6,11].

Face-selective patches of pSTS receive projections directly from the periphery of V1/V2[12], and through polysynaptic connections with intermediate stations that include the occipital face area (OFA) and middle temporal motion-processing area (V5/MT+)[9,13]. In turn, pSTS sends direct and indirect back-projections to V1/V2[14,15]. Consistent with these reciprocal connections, pSTS and V1/V2 also show intrinsic

[1]Centro studi e ricerche in Neuroscienze Cognitive, Dipartimento di Psicologia "Renzo Canestrari", Alma Mater Studiorum Università di Bologna, Cesena Campus, Cesena, Italy. [2]Neuroscience Area, International School for Advanced Studies (SISSA), Trieste, Italy. [3]IRCCS San Camillo Hospital, Venice, Italy. [4]Facultad de Lenguas y Educación, Universidad Antonio de Nebrija, Madrid 28015, Spain. [5]Dipartimento di Psicologia, Università degli Studi di Torino, Torino, Italy. [6]Department of Medical and Clinical Psychology, Tilburg University, Tilburg, The Netherlands. [7]Centro de Investigación en Neuropsicología y Neurociencias Cognitivas, Universidad Católica del Maule, Talca, Chile. ✉e-mail: sara.borgomaneri@unibo.it; m.tamietto@tilburguniversity.edu; alessio.avenanti@unibo.it

functional connectivity at rest[16,17]. This functional coupling is predictive of inter-individual differences in emotion recognition accuracy[18], and increases during the perception of emotional expressions[19]. However, a causal explanation of how pSTS and V1/V2 coordinate to support efficient emotion perception remains elusive and requires characterization of the network along several dimensions.

In this context, two critical elements of qualification concern the directionality and timing of information flow. Rather than a feedforward readout and linear integration of visual information along the cortical hierarchy, current models of visual awareness and perceptual decision-making assign increasing relevance to reentrant projections and recursive processing as general principles of visual recognition[20–22]. Accordingly, information transmission is reciprocal between adjacent stages and, in most cases, backward projections broadly outnumber forward projections[23]. Transcranial magnetic stimulation (TMS) has proven to be an ideal tool for probing the timing and function of feedback activity in the visual system. In fact, TMS can assess the causal impact of regional cortical activity on specific perceptual functions with millisecond precision. Moreover, because the neural activity induced by TMS spreads to anatomically connected regions[24,25], cortico-cortical information flow can be traced and temporal dynamics investigated. For example, seminal TMS studies have targeted back-projections from V5 to V1 in an early time window of approximately 40 ms and have demonstrated that they are necessary for visual awareness of motions[26,27].

Another avenue of inquiry concerns the malleability of these reentrant connections and whether plasticity can be recruited to improve visual perception. A TMS protocol, named cortico-cortical paired associative stimulation (ccPAS), can effectively strengthen synaptic connections and induce Hebbian plasticity that critically depends on both the direction and the timing of connectivity[28–37]. The ccPAS protocol involves the repeated pairing of TMS pulses over two brain areas with an interstimulus interval (ISI) consistent with the propagation of signals from the 'pre-synaptic' to the 'post-synaptic' target nodes[31–34]. This stimulation determines spike timing-dependent plasticity (STDP)[38–40] that is associated with changes in the strength of effective cortico-cortical connectivity between targeted areas, as shown by physiological assays addressing motor areas[31–36].

Notably, recent studies have applied ccPAS to the visual system, showing that strengthening reentrant connections from V5/MT to V1/V2 with an optimal timing of 20 ms between pulses has a transient impact on perceptual judgments, as it leads to enhanced detection of motion coherence, evident between 30 and 60 minutes after the stimulation[28–30].

However, there is currently no evidence that similar short-term plastic changes can be induced in brain areas like pSTS that are traditionally assigned to the ventral visual stream. Moreover, pSTS has been recently proposed as the terminal site of a third temporo-occipital pathway specialized for social perception[9], encompassing projections from early visual cortex (V1/V2) via motion-selective areas (MT+/V5)[9,12–15]. To address this issue, in the present study we tested the relevance and functional selectivity of back-projections from pSTS to V1/V2 in the perception facial expressions.

In a series of experiments, we provided causal evidence that 1) reentrant projections from pSTS to V1/V2 are functionally malleable with ccPAS; 2) exogenous strengthening of back-projections boosts sensitivity to facial expressions under noisy and difficult perceptual conditions; 3) this behavioral effect induced by ccPAS dovetails with enhanced electrophysiological activity in the pSTS-V1/V2 network in response to facial expressions, with maximal activity over V1/V2; 4) these plastic changes critically depend on the directionality and physiologically-defined timing of brain connectivity; and 5) they do not extend to other perceptual judgments, such as perception of gender under identical experimental conditions.

Experiment 1 combined TMS and EEG to assess the temporal profile of signal propagations from pSTS to V1/V2[24,25], and identified 200 ms as the optimal timing to mimic STDP and respect the Hebbian principle of consequentiality[38–40]. Experiment 2 exploited this knowledge to devise a time-resolved ccPAS protocol tailored for pSTS-to-V1/V2 reentrant connections. We showed that transient enhancement of emotion perception is contingent upon a stimulation interval of 200 ms between the two TMS pulses, as it disappeared with different intervals or when pSTS and V1/V2 were stimulated synchronously. Experiment 3 addressed direction-specificity and showed that improvements in emotion recognition do not occur when feedforward connections between V1/V2 and pSTS are stimulated, or when sham stimulation is delivered. Experiment 4 tested functional specificity, applying the same ccPAS protocol with a control task matched for difficulty and requiring participants to discriminate gender instead of facial expressions. Finally, Experiment 5 measured event-related potentials (ERPs) to examine the electrophysiological correlates of improved perception following the critical ccPAS manipulation. After ccPAS, early ERPs elicited by facial expressions (i.e., the P1 component) were enhanced in amplitude. Consistent with the Hebbian principle, the neuronal generators of this enhanced P1 amplitude were maximally expressed over the V1/V2 site, where TMS activations converged due to ccPAS targeting of pSTS-to-V1/V2 projections.

## Results

A total of 155 healthy young adults were recruited in 5 experiments and randomly assigned to 11 groups according to the specific TMS protocol administered and the task they were asked to perform.

### Experiment 1: Tracking signal propagation from pSTS to V1/V2

A first TMS-EEG co-registration study was designed to track signal propagation from pSTS to V1/V2 and estimate its timing. To this aim, we administered active and sham single-pulse TMS over the right pSTS in a group of 10 participants while EEG signals were continuously recorded. We analyzed the time-course of TMS-evoked responses at the sensor level (occipital electrodes O1, Oz, O2) and the source level in a region of interest (ROI) centered over the right occipital pole, corresponding to the location of V1/V2. Following pSTS stimulation, the maximal EEG peak of TMS-evoked activity was recorded from occipital electrodes (Fig. 1a) and the V1/V2 ROI (Fig. 1b) after ~200 ms, consistent with the recruitment of long-range and polysynaptic reentrant temporo-occipital connections.

Smaller and short-lasting activations were also observed in both V1/V2 and pSTS after ~100 ms, but they were not temporally specific or clearly distinct from activations observed over the stimulated pSTS itself (Supplementary Fig. 1). Based on these findings, in Experiments 2–5 we selected 200 ms as the critical ISI for targeting pSTS-to-V1/V2 back-projections, thus devising a novel long-latency(200 ms) ccPAS protocol[36] (Supplementary Fig. 2). An ISI of 100 ms was used as a control for testing the protocol's temporal specificity.

### Experiment 2 – Time-specific activation of the pSTS-to-V1/V2 pathway enhances visual perception of emotions from facial stimuli

In Experiment 2, we tested whether a long-latency ccPAS protocol aimed at strengthening pSTS-to-V1/V2 back-projections modulates the ability to perceive emotional expressions under challenging and noisy perceptual conditions.

Forty-two participants were randomly assigned to three different stimulation groups, according to the temporal properties of the ccPAS protocol (Fig. 2). The Experimental group (Exp2$_{STS-V1}$), underwent a ccPAS protocol repeatedly activating the pathway connecting pSTS and V1/V2; this protocol involved the administration of 90 pairs of TMS pulses, with the first pulse of each pair targeting the right pSTS and the second pulse targeting V1/V2 after a 200-ms ISI. Based on Experiment

1, pSTS-V1/V2 stimulation with this ISI was expected to induce STDP in tempero-occipital back-projections. Two control ccPAS conditions similarly targeted pSTS and V1/V2 but the ISI between the pulses was manipulated to prevent induction of STDP. In the first control group (Ctrl$_{100ms}$), we set the ISI between pSTS and V1/V2 stimulation to 100 ms, whereas in the second control group (Ctrl$_{0ms}$), the TMS pulses were delivered simultaneously (i.e., at an ISI of 0 ms; see Methods for details).

Participants in all groups were asked to perform the same emotion perception task (Fig. 2a), whereby faces showing expressions associated with happiness and fear were briefly presented in a sandwich masking procedure at three different exposure durations: 17, 33, or 50 ms (Fig. 2b). The task was administered before undergoing the assigned ccPAS protocol (i.e., at baseline), immediately after the ccPAS (T0), and again at 20, 40, 60, and 80 min following the ccPAS procedure (T20-T80; Fig. 2c). A preliminary control analysis ensured that performance at baseline was comparable across all groups in Experiment 2, as well as in the other experiments (Supplementary Table 1, 2, and 3).

The ccPAS (Exp2$_{STS-V1}$, Ctrl$_{0ms}$, Ctrl$_{100ms}$) × Exposure time (17, 33, 50 ms) × Time from ccPAS (T0, T20, T40, T60, T80) ANOVA on baseline-corrected d′ values showed a non-significant main effect of ccPAS ($F_{2,39} = 2.71$; $p = 0.079$) and a significant ccPAS × Exposure time interaction ($F_{4,78} = 2.56$; $p = 0.045$; $\eta_p^2 = 0.12$; see Fig. 3).

Post-hoc analysis showed that expression recognition improved following the ccPAS protocol in the most difficult condition − i.e., when faces were briefly displayed for 17 ms − but only for participants assigned to the experimental group (Exp2$_{STS-V1}$).

The improvement observed in the experimental group at the 17-ms exposure time was greater than in any other condition (all $p \leq 0.044$; all Cohen's $d \geq 0.54$; black asterisks in Fig. 3); there were no other differences between groups or exposures (all $p \geq 0.11$). A further two-tailed t-test showed that d′ values for expression recognition at an exposure timing of 17 ms were statistically higher following Exp2$_{STS-V1}$ ccPAS compared to the pre-ccPAS baseline ($t_{13} = 4.39$; $p < 0.001$; Cohen's $d = 1.17$; red asterisks in Fig. 3). No other ANOVA main effects or interactions reached significance (all $F \leq 1.91$; all $p \geq 0.13$), including the 3-way interaction ($F_{8.4,163.0} = 1.05$; $p = 0.37$). This indicates that the increased sensitivity, contingent upon Exp2$_{STS-V1}$ ccPAS, was comparable across post-ccPAS time points and lasted for at least 80 min.

Finally, increased sensitivity to briefly presented emotional faces was not due to changes in decision criteria or speed/accuracy trade-offs, as we observed no effect of ccPAS on response bias (β) or response times (RTs) (Supplementary Table 4).

### Experiment 3−Direction-specific activation of the pSTS-to-V1/V2 pathway enhances visual perception of emotion from facial stimuli

Experiment 3 investigated the directional specificity of the neuro-stimulation protocol. Thirty-nine new participants performed the same task and underwent the same general procedure used in Experiment 2 and were evenly assigned to three groups based on the ccPAS protocol (Fig. 2). The Experimental group (Exp3$_{STS-V1}$) was subjected to the same ccPAS protocol described in Experiment 2, i.e., first pulse over pSTS and second pulse over V1/V2 at the critical

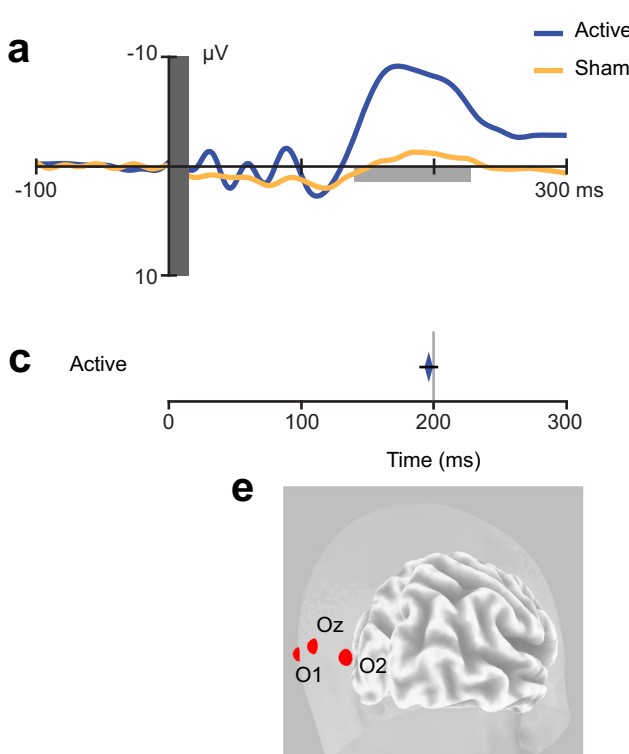

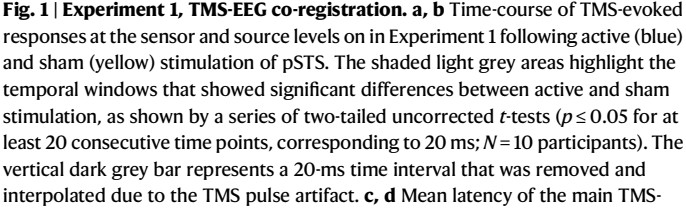

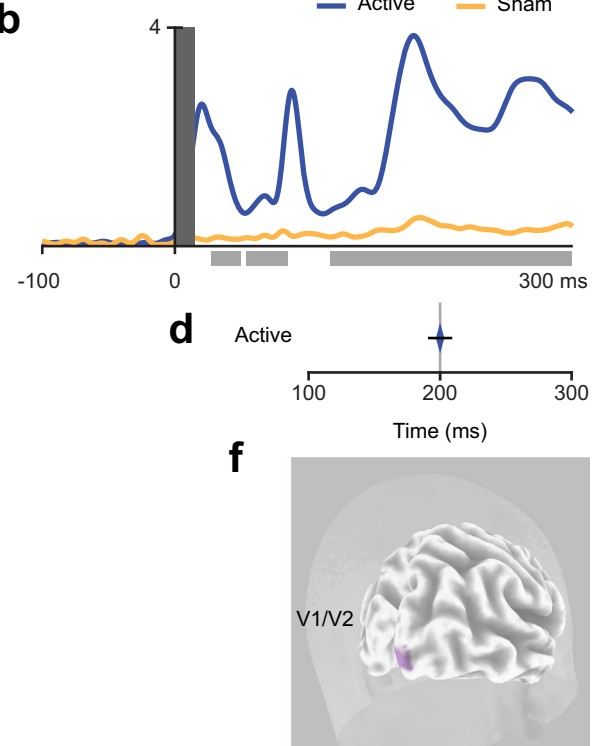

**Fig. 1 | Experiment 1, TMS-EEG co-registration. a, b** Time-course of TMS-evoked responses at the sensor and source levels on in Experiment 1 following active (blue) and sham (yellow) stimulation of pSTS. The shaded light grey areas highlight the temporal windows that showed significant differences between active and sham stimulation, as shown by a series of two-tailed uncorrected t-tests ($p \leq 0.05$ for at least 20 consecutive time points, corresponding to 20 ms; $N = 10$ participants). The vertical dark grey bar represents a 20-ms time interval that was removed and interpolated due to the TMS pulse artifact. **c, d** Mean latency of the main TMS-evoked components peaking at ~200 ms following pSTS stimulation at the sensor (197 ms; 95% confidence interval: [183 ms, 210 ms], **c**) and source (200 ms; 95% confidence interval: [183 ms, 216 ms], **d**) levels. The error bars represent the S.E.M. **e** Target sensors (O1, Oz, O2). **f** Cortical regions included in the V1/V2 ROI (Talairach coordinates: x = 19, y = −98, z = 1). Source data for latency results are provided as a Source Data file. Dataset to generate the EEG findings is provided at the following link: https://osf.io/yqbsj/.

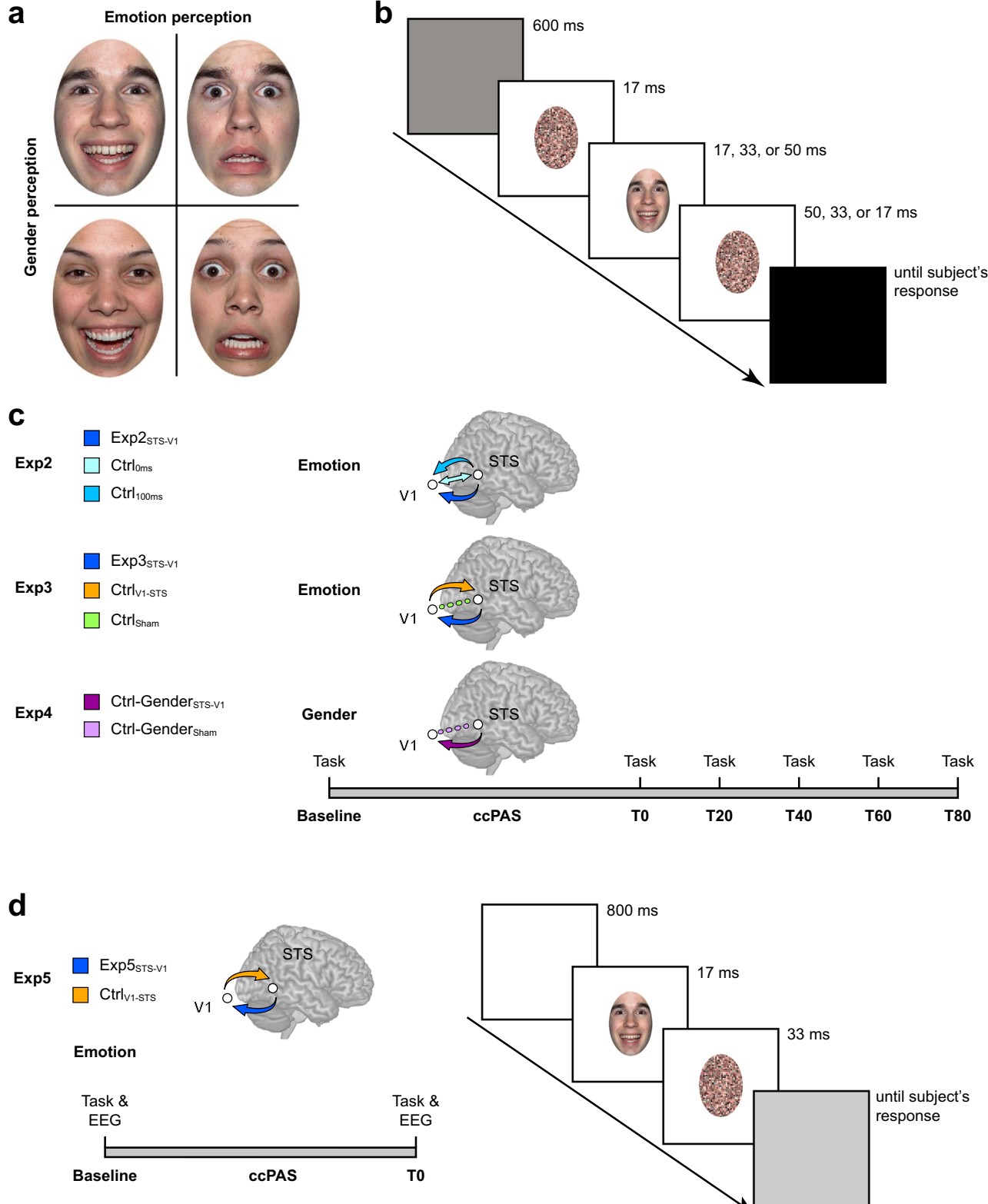

200-ms ISI. In the first control group (Ctrl$_{V1-STS}$), we reversed the order of the two TMS pulses: the first pulse of each TMS pair was delivered to V1/V2 and the second pulse to pSTS using the same 200-ms ISI, to potentially target feedforward connections from V1/V2 to pSTS. In a second control group (Ctrl$_{Sham}$), the ccPAS protocol was delivered using the same parameters as in the experimental condition but with the coil tilted at 90 degrees, thereby preventing the induction of currents in the brain.

The ccPAS (Exp3$_{STS-V1}$, Ctrl$_{V1-STS}$, Ctrl$_{Sham}$) × Exposure time (17, 33, 50 ms) × Time from ccPAS (T0, T20, T40, T60, and T80) ANOVA on baseline-corrected d′ values showed a main effect of Exposure time ($F_{1.7,61.9} = 4.32$; $p = 0.02$; $\eta p^2 = 0.11$) and, more importantly, a significant ccPAS × Exposure × Time interaction ($F_{3.4,61.9} = 2.64$; $p = 0.05$; $\eta_p^2 = 0.13$; Fig. 4).

We replicated the main results of Experiment 2, with a selective improvement in expression recognition only in the group subjected to

**Fig. 2 | Study design across Experiments 2–5. a** Examples of stimuli depicting male and female targets displaying happy and fearful expressions. The stimuli can be accessed at the following link: https://osf.io/yqbsj/. They have been adapted from the NimStim Set of Facial Expressions by Tottenham and colleagues[80], which was downloaded from https://danlab.psychology.columbia.edu/content/nimstim-set-facial-expressions. **b** Trial structure common to Experiments 2–4 with faces preceded and followed by scrambled images (sandwich masking). **c** ccPAS protocols and time course of the experimental session in Experiments 2–4. **d** ccPAS protocols, time course, and trial structure of Experiment 5 combining ccPAS and EEG. Face stimuli were backward-masked in Experiment 5 (instead of sandwich-

masked, as in the previous experiments) to avoid EEG activity due to the presentation of the first mask. Brain images in panel **c** and **d** are based on a template from the software MRIcron. Chris Rorden's MRIcron, all rights reserved. https://people.cas.sc.edu/rorden/mricron/install.html. Abbreviations in subscript indicate ccPAS conditions: STS-V1= ccPAS with first TMS pulse over STS and second pulse over V1/V2, and a 200-ms ISI; 0ms = ccPAS with simultaneous stimulation of STS and V1/V2 (0-ms ISI); 100ms = ccPAS with first pulse over STS and second pulse over V1/V2, and a 100-ms ISI; V1-STS = ccPAS with first pulse over V1/V2 and second pulse over STS, and a 200-ms ISI.

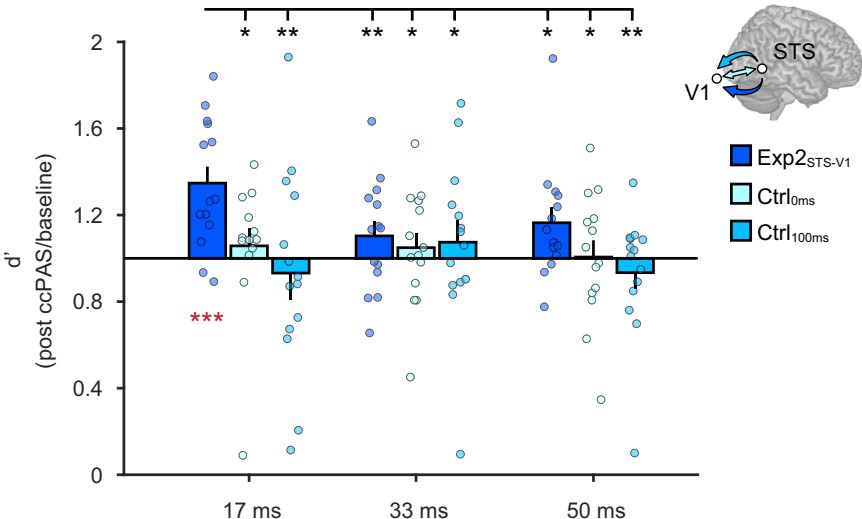

**Fig. 3 | Time-specific effects of ccPAS.** Experiment 2 results showing a selective improvement in visual perception of emotions when face stimuli are exposed for 17 ms. Histograms depict the significant ccPAS × Exposure time interaction ($F_{4,78} = 2.56$; $p = 0.045$; $\eta_p^2 = 0.12$; $N = 42$ participants), with mean d′ values across the post-ccPAS time points (average of T0, T20, T40, T60, T80) expressed relative to baseline values. Red asterisks indicate significant increase in d′ relative to baseline levels: following the critical ccPAS protocol that targets long-latency pSTS-to-V1/V2 backward connections using the ISI of 200 ms (i.e., Exp2$_{STS-V1}$), d′ values significantly increased relative to baseline, specifically in the 17-ms exposure condition ($t_{13} = 4.39$; $p = 0.0007$; *Cohen's d* $= 1.17$; $N = 14$ participants), with a mean

increase of +35% (95% confidence interval: [+19%, +50%]). The brain depicted in the figure is based on a template from the software MRIcron. Chris Rorden's MRIcron, all rights reserved. https://people.cas.sc.edu/rorden/mricron/install.html. Black asterisks denote significant post-hoc comparisons between this critical condition and other post-ccPAS conditions, using the Duncan test to correct for multiple comparisons (all $p \le 0.044$; all *Cohen's d* $\ge 0.54$). There were no changes following ccPAS protocols controlling for the timing of the paired stimulation with ISIs of 0 ms (Ctrl$_{0ms}$; $N = 14$ participants) or 100 ms (Ctrl$_{100ms}$; $N = 14$ participants). Dots represent individual data. Error bars denote S.E.M. All statistical tests are two-tailed. $^*p \le 0.05$, $^{**}p \le 0.01$, $^{***}p \le 0.001$. Source data are provided as a Source Data file.

pSTS-to-V1/V2 ccPAS at a 200-ms ISI (Exp3$_{STS-V1}$) and during short stimulus exposures (17 ms) relative to the other conditions (all $p \le 0.037$; all *Cohen's d* $\ge 0.67$; black asterisks in Fig. 4). Notably, post-hoc analyses showed no changes in face recognition when feedforward connections were targeted (Ctrl$_{V1-STS}$; all $p \ge 0.17$) or when sham stimulation was administered (Ctrl$_{Sham}$; all $p \ge 0.16$). A two-tailed t-test showed that d′ values for perception of facial expressions displayed for 17 ms were higher following Exp3$_{STS-V1}$ ccPAS compared to baseline levels ($t_{12} = 3.80$; $p = 0.003$; *Cohen's d* $= 1.05$; red asterisks in Fig. 4). No other main effects or interactions reached significance (all $F \le 1.55$; all $p \ge 0.17$).

Once again, in Experiment 3, changes in sensitivity were not due to shifts in decision criteria or to speed/accuracy trade-offs (Supplementary Table 4).

### Experiment 4 - Functional specificity of pSTS-to-V1/V2 back-projections

Experiment 4 tested the functional specificity of pSTS-to-V1/V2 ccPAS. We substituted the emotion perception task with a gender perception task, while keeping the stimuli and ccPAS protocols identical (Fig. 2a). As previously established[41–43], the gender perception task involves the ability to process morphological facial features and relies on ventral occipito-temporal face areas, rather than pSTS. Twenty-eight new participants were evenly assigned to two different groups. One gender

task group was subjected to the same active stimulation of pSTS-to-V1/V2 back-projections at a 200-ms ISI that proved effective at enhancing emotion perception in the previous experiments (Ctrl-Gender$_{STS-V1}$), while another control group received sham stimulation (Ctrl-Gender$_{Sham}$) (Fig. 2c).

We found no evidence that the same ccPAS protocol targeting pSTS-to-V1/V2 connections, which previously enhanced perception of emotions, also modulates perception of gender. In fact, the ccPAS condition (Ctrl-Gender$_{STS-V1}$, Ctrl-Gender$_{Sham}$) × Exposure time (17, 33, 50 ms) × Time from ccPAS (T0, T20, T40, T60, T80) ANOVA showed no significant main effects or interactions (all $F \le 2.31$; $p \ge 0.08$; Supplementary Table 4). In addition, we found no effect of ccPAS on β or RTs (Supplementary Table 4).

To further assess the functional specificity of pSTS-to-V1/V2 back-projections, we directly compared the three groups that received the same pSTS-to-V1/V2 ccPAS protocol but performed either the emotion perception task (in Experiments 2 and 3) or the gender perception task (in Experiment 4). The Experiment (Exp2$_{STS-V1}$, Exp3$_{STS-V1}$, Ctrl-Gender$_{STS-V1}$) × Exposure time (17, 33, 50 ms) × Time from ccPAS (T0, T20, T40, T60, T80) ANOVA on baseline-corrected d′ values showed significant main effects of Experiment ($F_{2,38} = 5.53$; $p = 0.008$; $\eta_p^2 = 0.23$) and Exposure time ($F_{2,76} = 11.27$; $p < 0.001$; $\eta_p^2 = 0.23$) and a significant Experiment × Exposure time interaction ($F_{4,76} = 3.16$; $p = 0.02$; $\eta_p^2 = 0.14$; Fig. 5). Importantly, d′ values increased at 17-ms

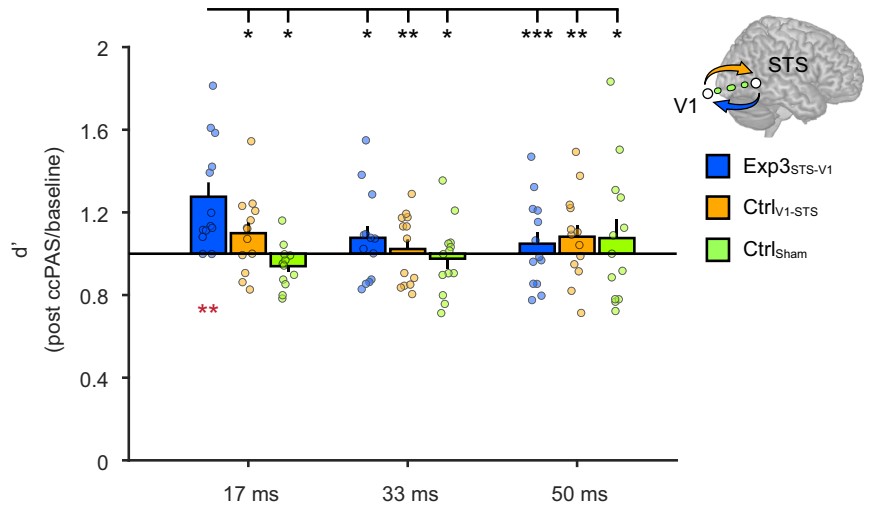

**Fig. 4 | Direction-specific effects of ccPAS.** Experiment 3 results replicating the selective improvement in visual perception of emotions when face stimuli are exposed for 17 ms. Histograms depict the significant ccPAS × Exposure time interaction ($F_{3.4,61.9} = 2.64$; $p = 0.05$; $\eta_p^2 = 0.13$; $N = 39$ participants), with mean d' values across the post-ccPAS time points (average of T0, T20, T40, T60, T80) expressed relative to baseline values. Red asterisks indicate significant increase in d' relative to baseline levels: as in Experiment 2, following the critical ccPAS protocol (Exp3$_{STS-V1}$), d' values significantly increased relative to baseline in the 17-ms exposure condition ($t_{12} = 3.80$; $p = 0.003$; *Cohen's d* = 1.05; $N = 13$ participants), with a mean increase of +28% (95% confidence interval: [+13%, +42%]). The brain depicted in the figure is based on a template from the software MRIcron. Chris Rorden's MRIcron, all rights reserved. https://people.cas.sc.edu/rorden/mricron/install.html. Black asterisks denote significant post-hoc comparisons between the critical condition and other post-ccPAS conditions, using the Duncan test to correct for multiple comparisons (all $p \le 0.037$; all *Cohen's d* $\ge 0.67$). There were no changes following ccPAS protocols controlling for the direction of connectivity (Ctrl$_{V1-STS}$; $N = 13$ participants) or nonspecific effects (Ctrl$_{Sham}$; $N = 13$ participants). Dots represent individual data. Error bars denote S.E.M. All statistical tests are two-tailed. $^*p \le 0.05$, $^{**}p \le 0.01$, $^{***}p \le 0.001$. Source data are provided as a Source Data file.

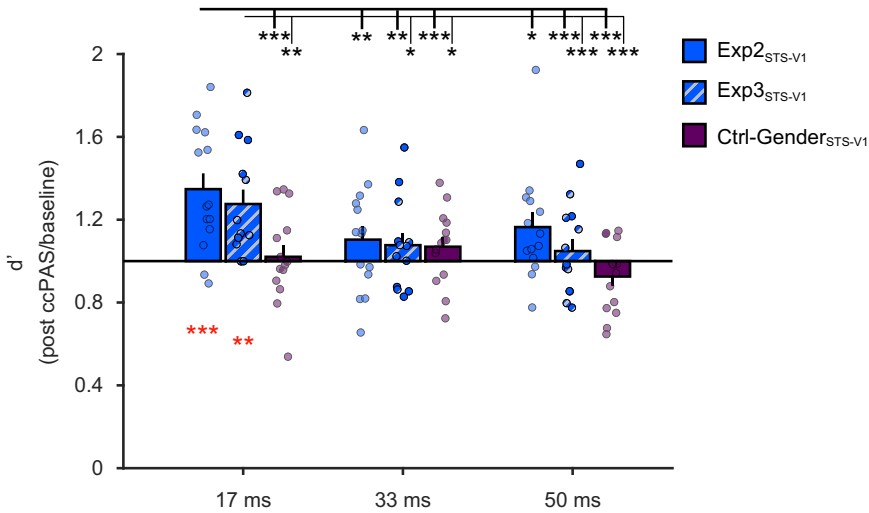

**Fig. 5 | Functional-specific effects of ccPAS.** Functional specificity of long-latency pSTS-to-V1/V2 ccPAS across Experiments 2-4. Histograms depict the significant ccPAS × Exposure time interaction ($F_{4,76} = 3.16$; $p = 0.02$; $\eta_p^2 = 0.14$; $N = 41$ participants), with mean d' values across the post-ccPAS time points (average of T0, T20, T40, T60, T80) expressed relative to baseline values. Red asterisks denote significant increase in d' relative to baseline levels: following the two critical ccPAS protocols (Exp2$_{STS-V1}$, Exp3$_{STS-V1}$), d' values significantly increased relative to baseline in the 17-ms exposure condition (Exp2$_{STS-V1}$: +35%; 95% confidence interval: [+19%, +50%]; $t_{13} = 4.39$; $p = 0.0007$; *Cohen's d* = 1.17; $N = 14$ participants; Exp3$_{STS-V1}$: +28%; 95% confidence interval: [+13%, +42%]; $t_{12} = 3.80$; $p = 0.003$; *Cohen's d* = 1.05; $N = 13$ participants). Black asterisks denote significant post-hoc comparisons between these critical conditions and other post-ccPAS conditions using the Duncan test to correct for multiple comparisons (all $p \le 0.034$; all *Cohen's d* $\ge 0.67$). There were no changes in gender perception (Ctrl-Gender$_{STS-V1}$, $N = 14$ participants). Dots represent individual data. Error bars denote S.E.M. All statistical tests are two-tailed. $^*p \le 0.05$; $^{**}p \le 0.01$; $^{***}p \le 0.001$. Source data are provided as a Source Data file.

exposure times for emotion perception, but not for gender perception under identical neurostimulation and visual presentation conditions. In both Exp2$_{STS-V1}$ and Exp3$_{STS-V1}$, d' values for emotion perception increased at 17-ms exposure times compared to the other exposure durations (all $p \le 0.034$; all *Cohen's d* $\ge 0.54$; black asterisks in Fig. 5); moreover, these increases in d' were larger than the comparable values for gender perception in the Ctrl-Gender$_{STS-V1}$ group (all $p \le 0.03$; all *Cohen's d* $\ge 0.86$). The increase in d' for emotion perception was comparable across Experiments 2 and 3 ($p = 0.41$). No other effects were observed (all $F \le 1.26$; all $p \ge 0.27$).

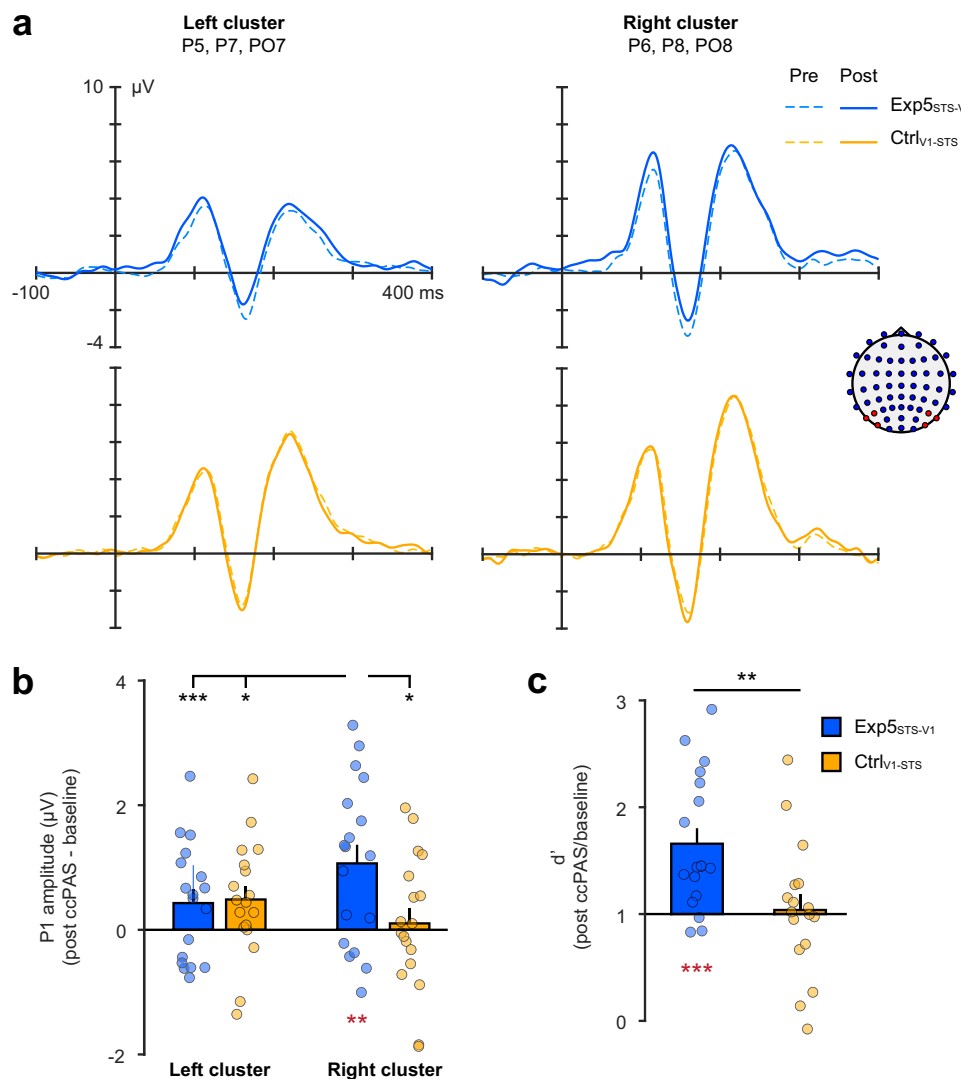

**Fig. 6 | Neural markers of improved perception following ccPAS.** Electro-physiological and behavioral changes following long-latency pSTS-to-V1/V2 ccPAS in Experiment 5. **a** Electrode clusters in the left unstimulated (PO7, P7, P5) and right stimulated hemispheres (PO8, P8, P6) showing grand average ERP waveforms pre-ccPAS (dashed line) and post-ccPAS (T0, continuous line). On the right side of the panel, the schematic representation of EEG electrode placements highlights the two electrode clusters in red, while depicting the other electrodes in blue. **b** Histograms depict the significant ccPAS × Electrode interaction ($F_{1,34} = 7.24$; $p = 0.01$; $\eta_p^2 = 0.18$; $N = 36$ participants), with changes in mean P1 amplitudes post-ccPAS (T0) expressed relative to baseline. Red asterisks denote a significant P1 increase relative to baseline (+1.27 μV; 95% confidence interval: [+0.61 μV, +1.94 μV]; $t_{17} = 3.77$; $p = 0.002$; *Cohen's d* = 0.89; $N = 18$ participants) following the critical ccPAS condition (Exp5STS-V1), specifically in the right cluster. Black asterisks denote significant comparisons between the increased P1 in the right cluster and the other

conditions using the Duncan test to correct for multiple comparisons (all $p \leq 0.033$; all *Cohen's d* ≥ 0.69). There were no changes following ccPAS protocols controlling for the direction of connectivity (CtrlV1-STS; $N = 18$ participants). **c** Histograms depict the significant effect of ccPAS ($F_{1,33} = 7.64$; $p = 0.009$; $\eta_p^2 = 0.19$), with mean d' values post-ccPAS (T0) expressed relative to baseline values and black asterisks indicating the d' difference between ccPAS protocols. Red asterisks denote a significant increase in d' relative to baseline levels (+66%; 95% confidence interval: [+37%, +95%]; $t_{17} = 4.41$; $p < 0.001$; *Cohens' d* = 1.04) following the critical ccPAS protocol (Exp5STS-V1; $N = 18$ participants). There were no changes following CtrlV1-STS ($N = 17$ participants). In panel **b** and **c**, dots represent individual data. Error bars denote S.E.M. All statistical tests are two-tailed. *$p \leq 0.05$, **$p \leq 0.01$, ***$p \leq 0.001$. Source data are provided as a Source Data file. Dataset to generate the EEG findings is provided at the following link: https://osf.io/yqbsj/.

## Experiment 5 - Electrophysiological correlates of improved perception following activation of the pSTS-to-V1/V2 pathway with ccPAS

Experiment 5 integrated ERPs to investigate the electrophysiological correlates of improved visual perception of emotions induced by long-latency ccPAS (Figs. 2d and 6a). Thirty-six new participants were randomly assigned to the Experimental group (Exp5STS-V1) targeting pSTS-to-V1/V2 back-projections with the critical 200-ms ISI, or a Control group (CtrlV1-STS) in which we reversed the order of the two TMS pulses, as in Experiment 3. To prevent EEG activity due to the initial presentation of scrambled picture, the first projected image was a face (i.e., sandwich masking was simplified to backward masking; Fig. 2d).

Moreover, we concentrated on the most relevant conditions: face-evoked ERPs were recorded during an emotion perception task at baseline (pre-ccPAS) and right after ccPAS (T0), and for the shortest stimulus exposure (17 ms), which was the only presentation condition previously found to be affected by ccPAS. All other aspects of the behavioral task remained identical to the previous experiments.

Figure 6 shows the results of Experiment 5. Behavioral findings replicated the perceptual improvement observed in prior experiments. Indeed, an ANOVA with the factor ccPAS (Exp5STS-V1, CtrlV1-STS) on baseline-corrected d' values showed a greater increase in Exp5STS-V1 compared to CtrlV1-STS ($F_{1,33} = 7.64$; $p = 0.009$; $\eta_p^2 = 0.19$; black asterisks in Fig. 6c). We also observed significantly higher d' values following

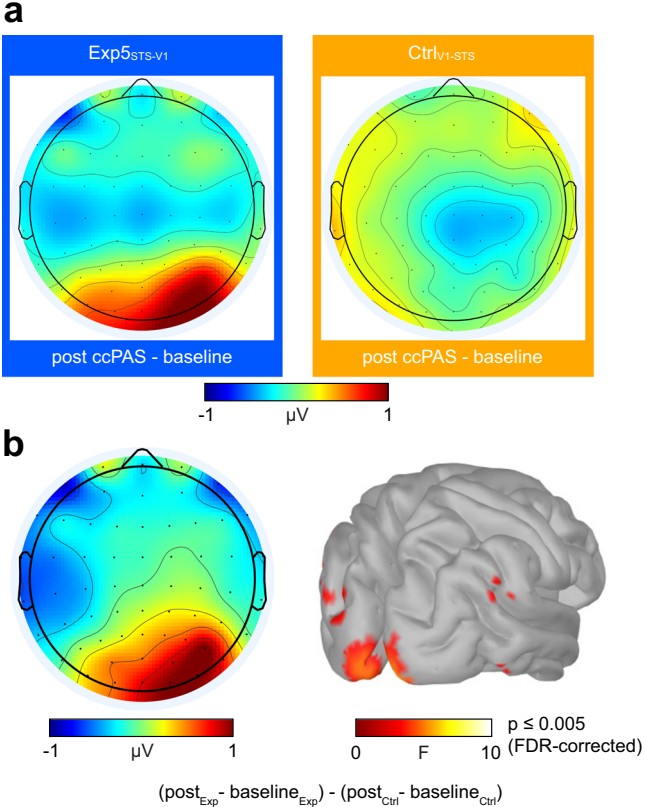

**Fig. 7 | Neural sources underlying ccPAS-specific P1 enhancement.** Localization of physiological changes following long-latency pSTS-to-V1/V2 ccPAS in Experiment 5. **a** Scalp map distribution showing changes in P1 amplitude in the Exp5$_{STS-V1}$ and Ctrl$_{V1-STS}$ groups following ccPAS (T0 vs. baseline). **b** Scalp distribution and source reconstruction showing higher increases in activity in the Exp5$_{STS-V1}$ group compared to the Ctrl$_{V1-STS}$ group (FDR-corrected $p \leq 0.005$). The increase in activity was maximal over early occipital regions, with the peak of activation at Talairach coordinates $x = 12$, $y = -98$, $z = -3$. All statistical tests are two-tailed. Dataset to generate the EEG findings is provided at the following link: https://osf.io/yqbsj/.

Exp5$_{STS-V1}$ ccPAS compared to pre-ccPAS baseline levels ($t_{17} = 4.41$; $p < 0.001$; Cohen's $d = 1.04$; red asterisks in Fig. 6c). See Supplementary Table 5 for RTs and β.

We investigated the influence of ccPAS on three key ERP components (P1, N170, P2) recorded over the right (stimulated) and the left (unstimulated) hemispheres (Fig. 6a; Supplementary Table 6). The ccPAS selectively affected the P1 component recorded over the targeted hemisphere. A ccPAS (Exp5$_{STS-V1}$, Ctrl$_{V1-STS}$) x Electrode cluster (Left, Right) ANOVA on baseline-corrected P1 amplitudes showed a main effect of Electrode cluster ($F_{1,34} = 10.53$; $p = 0.003$; $\eta_p^2 = 0.18$), no main effect of ccPAS ($F_{1,34} = 0.63$; $p = 0.43$), but, importantly, a ccPAS × Electrode interaction ($F_{1,34} = 7.24$; $p = 0.01$; $\eta_p^2 = 0.18$; Fig. 6b). Following Exp5$_{STS-V1}$ ccPAS, we observed an enhancement of P1 amplitudes over the right electrode cluster (PO8, P8, P6) relative to the other conditions (all $p \leq 0.03$; all Cohen's $d \geq 0.69$; black asterisks in Fig. 6b). No modulation was observed in the Ctrl$_{V1-STS}$ group between clusters ($p = 0.70$). A two-tailed $t$-test showed that P1 amplitudes in the right cluster – corresponding to the targeted hemisphere during ccPAS – were higher following Exp5$_{STS-V1}$ ccPAS compared to baseline levels ($t_{17} = 3.77$; $p = 0.002$; Cohen's $d = 0.89$; red asterisks in Fig. 6b).

Scalp maps show that the ERP enhancement in the P1 time window following Exp5$_{STS-V1}$ ccPAS, but not Ctrl$_{V1-STS}$, was mostly localized over right posterior electrodes (Fig. 7a). Source estimation in the P1 time window revealed that the neuronal generator of the effect was mostly localized in occipital cortices (with the peak of activation at Talairach coordinates $x = 12$, $y = -98$, $z = -3$, i.e., overlapping with the V1/V2 site

targeted during ccPAS), but also included a source compatible with the location of pSTS, with significantly higher activations for Exp5$_{STS-V1}$ than Ctrl$_{V1-STS}$ (Fig. 7b).

The ccPAS × Cluster ANOVAs on baseline-corrected N170 and P2 amplitudes showed no significant effects (all $F \leq 1.20$; all $p \geq 0.28$), suggesting that ccPAS selectively influenced early (P1) but not later ERP components (see Supplementary Table 6).

## Discussion

How the human visual system represents emotional signals and coordinates behavioral responses is a thriving topic in neuroscience. The present study delineates the features that permit short-term synaptic strengthening of reentrant connections from pSTS to V1/V2, leading to enhanced perception of facial expressions.

Prior work has suggested a functional coupling between pSTS and V1/V2 at rest and during emotion perception[16–19]. Although growing evidence suggests a key role for reentrant networks in visual awareness and perceptual decision-making[20–23,26–30], there has been no prior attempt to use ccPAS to modulate back-projections between critical visual nodes within a network for emotion processing. Indeed, causal evidence supporting the effects of recursive processing in the human visual system was limited to the involvement of the V5/MT-to-V1 pathway in low-level motion perception[26–30]. Moreover, prior ccPAS investigations into the role of back-projections have focused on perceptual performance[28–30], leaving it unclear whether behavioral enhancements are mediated by neurophysiological changes in the targeted networks.

Here we found that exogenous manipulation of pre- and post-synaptic nodes in the pSTS-to-V1/V2 pathway, in accordance with the temporal parameters of Hebbian plasticity, leads to plastic changes that boost sensitivity to facial expressions for at least 80 min. This effect was consistently observed across three independent experiments (Exp$_{STS-V1}$ groups in Experiments 2, 3 and 5) and occurred only when the temporal features of the ccPAS protocol matched those estimated to be optimal from the TMS-evoked EEG response in Experiment 1 (i.e., with a 200-ms ISI; Supplementary Fig. 2). The enfolding and long-lasting behavioral effect of ccPAS (see Supplementary Fig. 3) appears in keeping with prior TMS studies on Hebbian plasticity in humans[28,33–36]. In fact, visual sensitivity was increased immediately after the stimulation protocol and persisted for at least 80 min, following an inverse U-shaped curve peaking at around 40–60 min (Supplementary Fig. 3).

The improved ability to recognize emotions in observed facial expressions was paralleled by an increase in the P1 component amplitude post-ccPAS, reflecting strengthened temporo-occipital activations during early stages of visual processing. This electrophysiological biomarker, though measurable in pSTS, was maximal over V1/V2 where neural activations were expected to converge due to ccPAS targeting of pSTS-to-V1/V2 back-projections. In keeping with this idea, the neuronal generators of the enhanced P1 activity nicely corresponded to the coordinates of the stimulated sites during ccPAS and were not found following stimulation of the same sites in forward direction (V1/V2-to-pSTS).

Perceptual enhancement did not take place if the requirements of STDP in the pSTS-to-V1/V2 pathway were not met. In fact, none of the participants in the conditions controlling for timing (Ctrl$_{0ms}$ and Ctrl$_{100ms}$), directionality (Ctrl$_{V1-STS}$) or nonspecific effects (Ctrl$_{Sham}$) improved in emotion perception following ccPAS administration. Compelling evidence indicates that perceptual decisions arise as the consequence of recursive loops in the visual system, where initial activity in early areas is "explained away" by backward-flowing information from structures at subsequent processing stages, until the visual representation stabilizes and recognition is achieved[20,21,44]. Recurrent processing seems particularly important when stimuli are degraded, noisy, or otherwise ambiguous[45,46]. As it happens, initial

responses to degraded stimuli are weakened in the visual system but may be re-instantiated by recurrent processing[47–49]. Our results are in keeping with this notion; in fact, ccPAS improved emotion perception only when faces were briefly presented (17 ms) under noisy (masking) conditions, whereas no significant variation in perceptual sensitivity was observed at longer exposures.

One may wonder why no changes in performance or electrophysiological responses were detected following ccPAS in the $Ctrl_{V1-STS}$ groups of Experiments 3 and 5. In principle, reversing the order of the ccPAS pulses (i.e., first TMS pulse over V1/V2, second pulse over pSTS) should strengthen feedforward connections in the network. However, the lack of reliable electrophysiological changes following $Ctrl_{V1-STS}$ ccPAS suggests that this protocol was less effective than $Exp_{STS-V1}$ ccPAS in modulating cortico-cortical networks. Notably, the ISI of the ccPAS protocol was selected based on TMS-EEG co-registration[24,25] aimed at detecting remote effects of pSTS stimulation over early visual cortices, which yielded maximal activity in V1/V2 after ~200 ms. Here we have demonstrated that this approach can guide exogenous manipulation of long-latency reentrant connectivity from pSTS to V1/V2. Future studies could use a similar TMS-EEG approach (i.e., investigating remote effects of V1/V2 stimulation over higher-order visual areas) to develop and test feedforward-specific ccPAS manipulations.

The present study also addressed the functional specificity of the long-latency ccPAS procedure. Participants in the $Exp2_{STS-V1}$, $Exp3_{STS-V1}$ and $Exp5_{STS-V1}$ groups showed a consistent improvement in emotion perception, whereas no effect on gender perception was found under identical conditions in Experiment 4, even though the two tasks were matched for difficulty.

This enhanced sensitivity to emotional expressions cannot be attributed to changes in response bias, nonspecific effects, or speed/accuracy trade-offs (Supplementary Table 4 and 5). Neural populations in sectors of pSTS respond specifically to emotional expressions, as convincingly demonstrated by single-cell recordings in monkeys[50] and humans[51,52]. Likewise, pSTS shows greater activation when participants are asked to recognize facial expressions[53,54] than when they are asked to recognize faces based on morphological cues. Indeed, visual recognition of facial identity, age, or gender relies more on ventral occipito-temporal face-selective areas[41–43]. Accordingly, repetitive TMS studies have provided causal evidence for a dissociation between OFA and pSTS in discriminating faces based on morphological vs. expressive features, respectively[55–59]. Our study expands these prior findings by providing evidence of a functionally specific cortico-cortical neural mechanism through which pSTS can contribute to accurate perception of emotional facial expressions via top-down influence over early visual cortex.

Evidence from non-human primates shows that the dominant direction of signal flow within occipito-temporal networks is feedforward shortly after stimulus onset (i.e., within 150–200 ms), but then gradually reverses to feedback at longer latencies and remains so at rest[20,60]. The long-latency 200-ms ISI of the critical ccPAS protocol is in keeping with these neural dynamics and suggests that its efficacy depends upon the relative delay of backward communication within temporo-occipital networks[60,61]. We cannot rule out the possibility that the ccPAS protocol with a 200-ms ISI also indirectly recruited larger networks besides occipito-temporal areas. Indeed, TMS effects can be site- and function-specific, but not necessarily site-limited[24,25]. Yet, we did not find support for this possibility, as the EEG results in Experiment 5 located the effects of ccPAS within the stimulated sites of the occipito-temporal network, primarily in V1/V2, as predicted by Hebbian STDP.

Rapid feedback interactions coexist with the initial and dominant feedforward signal flow and can influence basic levels of visual processing[60–63]. Likewise, early interference with pSTS through rTMS in the 60–140-ms time window impairs ongoing perception of facial expressions[55]. These findings suggest that pSTS exerts its influence on emotion processing at multiple stages. This does not contradict our results. In fact, the early influence of pSTS on perception was reported during a single site, online rTMS interference protocol in an emotional face recognition task[55], whereas the long latency of our ccPAS protocol referred to the time window of maximal pSTS-to-V1/V2 interactions evidenced by single pulse TMS-EEG co-registration at rest (Experiment 1). The ccPAS protocol itself was administered offline in Experiments 2–5, and its behavioral impact was measured in a subsequent task. Further evidence of multiple temporal windows of pSTS influence also comes from Experiment 1, where we detected faster (although smaller and short-lasting) effects of pSTS stimulation on activity in early visual areas. These findings may inform further ccPAS manipulations based on rapid pSTS-V1/V2 feedback interactions, and future research could leverage both short- and long-latency neural interactions to promote Hebbian plasticity.

In conclusion, our study demonstrated that ccPAS aimed at strengthening the synaptic efficacy of long-latency pSTS-to-V1/V2 connections selectively enhances visual sensitivity to facial expressions. We provided the first causal evidence that pSTS-to-V1/V2 connections are malleable and afford a neural mechanism functionally relevant to emotion recognition. Furthermore, plastic enhancement critically depended on a time-resolved pairing of pre-and post-synaptic nodes that mimics STDP of temporo-occipital interactions. Our study thus provides proof of principle that long-latency ccPAS can be used to improve visual functions in healthy humans. These findings have theoretical and methodological implications, as they suggest that ccPAS can target complex cortico-cortical pathways while maintaining functional specificity. Moreover, we add to the growing literature showing the potential utility of non-invasive brain stimulation for improving cortical functions in humans[28,35,64–67].

## Methods
### Participants
A total of 155 healthy young adults were involved in the study. In Experiment 1, 10 participants (6 females and 4 males; mean age ± standard deviation: 22.1 y ± 2.2) were tested using TMS-EEG co-registration. In Experiment 2, 42 participants (22 females and 20 males; 23.9 y ± 2.2) were randomly assigned to one of three ccPAS conditions ($Exp2_{STS-V1}$, $Ctrl_{0ms}$, $Ctrl_{100ms}$) testing the temporal specificity of backward connectivity. In Experiment 3, 32 participants (15 females and 17 males; 23.6 y ± 2.8) were randomly assigned to one of three ccPAS conditions ($Exp3_{STS-V1}$, $Ctrl_{V1-STS}$, $Ctrl_{Sham}$) testing for directional specificity and nonspecific effects. In Experiment 4, 28 participants (19 females and 9 males; 22.8 y ± 2.5) were randomly assigned to one of two conditions ($Ctrl$-$Gender_{STS-V1}$, $Ctrl$-$Gender_{Sham}$) testing for task specificity. Finally, in experiment 5, 36 participants (15 females and 21 males; 22.9 y ± 2.6) were randomly assigned to one of two ccPAS conditions ($Exp5_{STS-V1}$, $Ctrl_{V1-STS}$) testing for directional specificity using behavioral and ERP methods. Participants were recruited through a combination of printed and electronic advertisements displayed on notice boards at different University of Bologna sites, as well as through word of mouth. Four additional participants were tested in the initial phases of Experiments 2–4 but excluded because of technical failures, either before or during ccPAS administration. In Experiment 5, baseline behavioral data from one participant in the control group were lost due to a technical failure, so this participant was excluded from analyses of behavioral data, but included for EEG data analyses. No participant was tested in more than one experiment.

We chose the sample size of Experiment 1 based on prior TMS-EEG work investigating TMS-evoked responses[68–70]. We estimated the sample of the experimental groups in Experiments 2 and 3 based on prior work in our lab investigating the effect of V5-V1 ccPAS on motion perception[28,29] and the effect of STS-rTMS on emotion perception[57], all showing large effect sizes (mean *Cohen's d* = 1.10). Using G*Power 3 software[71] with power (1–β) = 0.95 and α = 0.05, we estimated that a

sample of 11 participants would be sufficient to show baseline vs. post-ccPAS differences in the experimental groups. We decided to slightly increase this sample to 13/14 participants for each experimental or control group in Experiments 2–4. Moreover, we increased the sample to 18 participants for each group of Experiment 5, testing not only behavioral but also physiological data. The resulting sample sizes of all the experiments were similar to or greater than those of prior STS-rTMS studies on emotion perception[55–59].

All the participants were right-handed according to a standard handedness inventory[72], had normal or corrected-to-normal visual acuity in both eyes, and were naive as to the purposes of the experiment. None had neurological, psychiatric, or medical problems or any contraindication to TMS[73]. Participants provided written informed consent. The procedures were approved by the Bioethics Committee at the University of Bologna and were carried out in accordance with the ethical standards of the Declaration of Helsinki. No discomfort or adverse effects of TMS were reported or noticed during the experimental sessions.

## Experiment 1: TMS-EEG experiment

We used TMS-EEG co-registration to track the time-course of the pSTS influence over V1/V2 and thus identify a critical ISI for designing the ccPAS protocol we would use in Experiments 2–4. Participants received 60 active and 60 sham TMS pulses at rest over a right pSTS site that was identified using neuronavigation (see below). EEG signals were acquired with a TMS-compatible EEG amplifier (BrainAmp DC, BrainProducts GmbH, Germany) and 60 electrodes (EasyCap GmbH, Germany) mounted on an elastic cap according the standard 10/5 coordinate system. To monitor eye movements and blinks, three electrodes were placed on the outer canthi of both eyes and beneath the left eye. Reference and ground electrodes were placed on the right mastoid and AFz, respectively. The impedance was kept below 5 kΩ, and the electrode lead wires were arranged properly in order to reduce the TMS-induced electrical artifact[74]. EEG signals were digitized at a sampling rate of 5 kHz and low-pass filtered at 1 kHz (DC-recording). The analysis was performed using EEGLAB 2022.1[75] running on MATLAB. The fast-rising, fast-falling magnetic artifact and the early TMS-evoked muscle activity were removed by cutting and interpolating (cubic interpolation) the EEG signals in the interval from 1 ms before to 20 ms after TMS. A high-pass filter (Hamming windowed sinc FIR filter, cutoff frequency = 0.01 Hz) was then applied and signals were down-sampled to 1000 Hz. Continuous signals were segmented into a window (−100, 600 ms) around the TMS pulse and baseline-corrected to a time period of 90 ms (−100 to −10 ms) preceding TMS administration. EEG data were preprocessed to remove noisy epochs and correct muscular or eye artifacts with independent components analysis[76]. TMS-evoked responses were analyzed at the sensor and source level to identify activity peaks in V1/V2 following pSTS stimulation.

For sensor-level analyses, we averaged the signals from posterior occipital electrodes (O1, Oz, O2). For the source analysis, we estimated current source densities by projecting scalp potentials to source space using standardized low-resolution brain electromagnetic tomography (sLORETA - v20171101)[77–79]. TMS responses were projected onto a realistic head model based on the MNI152 template and restricted to cortical gray matter. A region of interest (ROI) approach was applied to measure the time-course of cortical responses in V1/V2. Specifically, a spherical ROI with a 20 mm radius was centered on the V1/V2 stimulation coordinates (V1-ROI) used in Experiments 2–4 (see the Neurostimulation paragraph below), and mean activity was extracted across the voxels contained within the ROI. In order to rule out possible contamination due to the spread of local activation in the TMS target area, mean activity after both active and sham TMS was extracted from a spherical ROI (20 mm radius) centered on pSTS coordinates (STS-ROI, see supplementary material).

## Experiments 2–5: general design

The experiments were programmed using MATLAB 2011b software to trigger TMS pulses, control stimulus presentation, and acquire behavioral responses. In each experiment, participants were randomly assigned to different groups according to the ccPAS protocol they would undergo. To test the effect of ccPAS on behavior, participants performed an emotion perception task (Experiments 2, 3 and 5) or a gender perception task (Experiment 4) before undergoing their assigned ccPAS protocol (i.e., at baseline), immediately after ccPAS administration (T0), and 20 (T20), 40 (T40), 60 (T60) and 80 (T80) minutes after ccPAS. In Experiment 5, participants performed the emotion perception task at baseline and T0 while we simultaneously recorded EEG activity.

For Experiments 2-5, we implemented a double-blind procedure: participants were blinded to group allocation, and the experimenters who collected and analyzed the data were blinded to the ccPAS conditions. The experimenters who administered ccPAS were not blinded to group allocation because they had to set TMS parameters (i.e., order of pulses, ISI, and orientation of the coils).

**Experiments 2, 3 and 4: ccPAS and behavior.** Pictures of faces displaying expressions associated with emotions were presented on a 19-inch screen located about 70 cm away from the participant. 16 happy and 16 fearful expressions from 16 models (8 females and 8 males) were selected from the Nimstim database[80] and adapted using Adobe Photoshop. Mirror-reflected copies of the faces were also created, so that the total number of stimuli was 64. Each face was cropped using an elliptical stencil to exclude hair, ears and neck so that we could rule out any effects of other physical components besides the facial expression[81] (Fig. 2a). Using a custom-made MATLAB script, we created mosaic pattern pictures made up of scrambled fragments of each face; we employed these stimuli as visual masks, each preserving the elliptical form, the color and the spatial frequency of the original picture[81,82].

In Experiments 2 and 3, participants performed a 2-alternative forced choice (2AFC) emotion perception task. On each trial, they were presented with a face and asked to discriminate the target's perceived emotional expression (forced choice: "happy" or "fearful"). In Experiment 4, participants were exposed to the same pictures but asked to perform a 2AFC gender perception task, requiring them to report the target's perceived gender (forced choice: "female" or "male").

The tasks were performed in blocks of 192 trials, including 3 sandwich-masked repetitions of the 64 face stimuli using 3 different exposure times. Each trial started with a gray screen (600 ms duration), followed by a forward masking stimulus (17 ms duration) that preceded the target face presented at the center of the screen (Fig. 2b). Faces were presented for 17, 33 or 50 ms, and then immediately replaced by a backward masking stimulus, which remained on the screen for 50, 33 or 17 ms respectively, to keep a constant stimulus duration of ~83 ms. A black screen was presented until the participants responded. Participants were provided their response by pressing one of two different keys on a keyboard with the index or middle finger of their right hand. They were asked to be as fast and as accurate as possible. Response–button correspondence was randomized across participants. Each block lasted approximately 5 min.

**Experiment 5: ccPAS, behavior and ERPs.** Participants performed a 2AFC emotion perception task as in Experiments 2–4 while EEG was simultaneously acquired. Visual stimuli consisted of pictures of faces displaying expressions associated with emotion presented on a 15-inch screen located about 60 cm from the participant. We used 10 happy and 10 fearful expressions from 10 models (5 females and 5 males) from the pool used in Experiments 2–4. Mirror-reflected copies of the faces were also created, so that the total number of stimuli was 40 and each face was presented three times, for a total of 120 trials. As shown

in Fig. 2d, each trial started with a white screen (800 ms duration), followed by the target face (17 ms duration) presented at the center of the screen and immediately replaced by a backward masking stimulus, which remained on the screen for 33 ms, to keep a constant stimulus duration of ~50 ms. A gray screen was presented until the participant's response.

EEG was acquired with the same EEG system and software as in Experiment 1. The signal was down-sampled to 500 Hz, low-pass filtered (cut-off frequency = 40 Hz, FIR filter), and re-referenced to the linked mastoid. Continuous signals were epoched in a window (−200, 600 ms) around the stimulus. Unique, non-stereotyped artifacts such as eye blinks were corrected using independent components analysis. Bad epochs (the ones presenting huge rubbing artifacts or undefined significant noise) were removed by visual inspection. ERP components (P1, N170, P2) were calculated separately for each channel and condition by selecting a 40-ms time window for the P1 (100–140 ms), N170 (150–190 ms) and P2 (200-240) components[83,84] and computing peak amplitudes. For sensor level analyses, we averaged the signals from occipito-parietal clusters of electrodes in the right (PO8, P8, P6) and left (PO7, P7, P5) hemispheres[85], after visual inspection of each component for each electrode without considering the condition to avoid circularity[86].

## ccPAS protocols

The ccPAS protocols were delivered with a Magstim BiStim2 machine (Magstim Company, UK) via two 50 mm figure-of-eight coils placed over the right pSTS and V1/V2. 90 pairs of stimuli were continuously delivered at a rate of 0.1 Hz for ~15 min[28–30,32,33,35–37], with each pair of stimuli consisting of two monophasic transcranial magnetic pulses. The pulses were triggered remotely using a computer that controlled both stimulators. TMS intensity was set to 60% of the maximum stimulator output[30]. The ccPAS protocol was manipulated in different groups of participants in Experiments 2–5.

## Experiment 2: Testing time-specific activation of backward connections

**Experimental condition: Exp2$_{STS-V1}$.** In each TMS pair, the first pulse was delivered to pSTS and followed by a second pulse delivered to V1/V2 with an ISI of 200 ms, in accordance with Experiment 1. This timing was critical to induce convergent activation of V1/V2 neurons via stimulation of pSTS and V1/V2 and thus induce STDP in pSTS-to-V1/V2 pathways[38–40]. The protocol was designed to strengthen reentrant connections from pSTS to V1/V2, thus enhancing the area of convergent activation, i.e., V1/V2 (Supplementary Fig. 2).

**Simultaneous active control for timing: Ctrl$_{0ms}$.** In this condition, both pulses were delivered simultaneously (ISI = 0 ms). According to the Hebbian principle[39,40], a synapse increases its efficacy if the presynaptic neuron persistently takes part in firing the post-synaptic target neuron. However, if two neurons fire at the same time, then one cannot have caused or taken part in firing the other. Thus, although I-wave interactions may occur during simultaneous TMS pulses[87], no net STDP is expected following Ctrl$_{0ms}$[28,30].

**Asynchronous active control for timing: Ctrl$_{100ms}$.** Stimulation was identical to that of the experimental condition except that pulses were delivered at a non-optimal ISI of 100 ms. Based on Experiment 1, the cortico-cortical volley elicited by pSTS stimulation (first pulse) would not consistently activate V1/V2 neurons at the time of exogenous V1/V2 stimulation (second pulse), thus failing to produce the convergent V1/V2 activation which is crucial for inducing STDP. This ccPAS condition controlled for timing-dependent effects. That is, it allowed us to verify that effects found in the Exp2$_{STS-V1}$ condition were timing dependent and not provoked by any consistent stimulation pairing the targeted areas.

## Experiment 3: Testing direction-specific activation of the pSTS-V1/V2 network

**Experimental condition: Exp3$_{STS-V1}$.** This pSTS-to-V1/V2 ccPAS group was identical to the Exp2$_{STS-V1}$ condition and aimed at replicating the effect observed in Experiment 2.

**Active control for direction: Ctrl$_{V1-STS}$.** In this condition we switched the direction of the associative pulses: the first pulse was given to V1/V2 and the second pulse to pSTS at the same ISI as the experimental condition (i.e., 200 ms). The Ctrl$_{V1-STS}$ group controlled for direction-dependent effects. That is, it allowed us to verify that any effect found in the experimental condition was the result of enforced feedback connections (from pSTS to V1/V2) and not found when reversing the order of the pulses, potentially activating feedforward connections.

**Sham control for nonspecific effects: Ctrl$_{Sham}$.** Stimulation in this sham condition was identical to that of the experimental condition, except for the fact that the TMS coils were tilted at 90 degrees, so that no current was induced in the brain throughout the ccPAS session.

## Experiment 4: Testing task specificity

**Active stimulation for control task: Ctrl-Gender$_{STS-V1}$.** The ccPAS protocol was identical to the experimental conditions of Experiments 2 and 3, but participants performed a gender perception task instead of the emotion perception task.

**Sham stimulation for control task: Ctrl$_{Sham}$.** Stimulation in this condition was identical to the Ctrl$_{Sham}$ condition of Experiment 3. Participants performed a gender perception task.

## Experiment 5: neurophysiological correlates of improved emotional expression perception

**Experimental condition: Exp5$_{STS-V1}$.** This pSTS-to-V1/V2 ccPAS protocol was identical to the Exp2$_{STS-V1}$ and Exp3$_{STS-V1}$ conditions and aimed at replicating the effects observed in Experiments 2 and 3.

**Active control for direction: Ctrl$_{V1-STS}$.** This V1/V2-to-pSTS ccPAS group was identical to the Exp3 Ctrl$_{V1-STS}$ condition and aimed at controlling for direction-dependent effects.

In both the experimental and control conditions, the behavioral task was adapted to EEG acquisition.

## Neuronavigation

In all experiments, the pSTS and V1/V2 sites were individually targeted using image-guided neuronavigation. The positions of the two coils were identified on each participant's scalp using the SofTaxic Navigator System (Electro Medical Systems) as in prior research[35–37,88–90]. Skull landmarks (nasion, inion, and 2 preauricular points) and ~100 points providing a uniform representation of the scalp were digitized by means of a Polaris Vicra digitizer (Northern Digital). An individual estimated magnetic resonance image (MRI) was obtained for each subject through a 3D warping procedure, fitting a high-resolution MRI template with the participant's scalp model and craniometric points. This procedure has been proven to ensure a global localization accuracy of roughly 5 mm[91].

Stimulation sites were identified in Talairach space on the basis of previous fMRI and TMS studies. When necessary, MNI coordinates were converted into Talairach space using GingerALE v. 2.3.1. The pSTS was localized in the right hemisphere at the coordinates x = 53, y = −49, z = 10, estimated by averaging subject-weighted coordinates identified in a recent meta-analysis[92] during emotion evaluation (75 experiments, 1742 participants) and passive observation of emotional facial expressions (20 experiments, 411 participants). The pSTS site is well in keeping with other brain imaging meta-analyses on emotional face perception[53,93] and prior TMS studies[56–58], falling within the range of

interindividual variability of the face-selective area in the pSTS reported by Sliwinska and Pitcher[59]. To localize V1/V2, we identified the scalp location that corresponded best to early visual cortex[94,95] (x = 19, y = −98, z = 1).

The pSTS and V1/V2 scalp locations identified by neuronavigation were marked with a pen on each participant's head and used to place the coils. Then, SofTaxic automatically estimated the individual Talairach coordinates corresponding to the projection of the targeted scalp sites onto the surface of the MRI-constructed stereotaxic template. These estimated coordinates indicated the most superficial cortical site where TMS effects were expected to be maximal. The mean coordinates (± standard deviation) of the targeted pSTS cortical site corresponded to the most posterior sector of Brodmann area 21 (Experiment 1: x = 56 ± 2, y = −50 ± 2, z = 9 ± 2; Experiment 2: x = 58 ± 3, y = −50 ± 2, z = 9 ± 2; Experiment 3: x = 59 ± 3, y = −49 ± 2, z = 9 ± 1; Experiment 4: x = 57 ± 2, y = −50 ± 2, z = 9 ± 1; Experiment 5: x = 57 ± 3, y = −49 ± 2, z = 9 ± 2). The mean coordinates of the targeted V1/V2 cortical site corresponded to Brodmann area 17 in the middle occipital gyrus (Experiment 2: x = 19 ± 1, y = −96 ± 1, z = 1 ± 1; Experiment 3: x = 19 ± 1, y = −96 ± 1, z = 1 ± 2; Experiment 4: x = 18 ± 1, y = −97 ± 1, z = 0 ± 1; Experiment 5: x = 18 ± 1, y = −97 ± 1, z = 0 ± 1). In Experiment 1, ROIs were centered over the searched coordinates (pSTS-ROI: x = 53, y = −49, z = 10; V1/V2-ROI: x = 19, y = −98, z = 1). Figures 2, 3, 4 and Supplementary Fig. 2 display schematic representations of the stimulated sites on a standard MRI template (Colin-27) from MRIcron.

### Data analysis

Behavioral data were processed offline. Response times (RTs) were calculated after removing trials with an incorrect (-13%) or slow (≥1 s) response (~4%). Accuracy was converted into measures of sensitivity (d') and response bias (β) in accordance with signal detection theory[96]. In the emotion (or gender) perception task, two types of responses were scored as correct: a "fearful" ("male") response to a fearful expression (male face) counted as a hit and a "happy" ("female") response to a happy expression (female face) counted as a correct rejection. Two types of responses were scored as incorrect: a "fearful" ("male") response to a happy expression (female face) counted as a false alarm and a "happy" ("female") response to a fearful expression (male face) counted as a miss.

To compare the effects of ccPAS across Experiments 2–4 and normalize the data distributions, changes in performance were baseline-corrected. d' and RT values at each post-ccPAS time point (T0, T20, T40, T60, T80) and for each exposure time (17, 33, 50 ms) were divided by the corresponding baseline values, whereas post-ccPAS response bias (β) values were baseline-corrected by subtracting baseline values. The same normalization was computed in Experiment 5 for post-ccPAS (T0) values relative to baseline values of d', RTs and β. In Experiments 2-4, mixed factors ANOVAs were performed on baseline-corrected d', β and RT values with ccPAS as a between-subjects factor and Exposure time (17, 33, 50 ms) and Time from ccPAS (T0, T20, T40, T60, T80) as within-subjects factors. Behavioral data in Experiment 5 were analyzed using a 1-way ANOVA with the between-subjects factor ccPAS. Electrophysiological data were analyzed offline. Mixed factors ANOVAs were performed on baseline-corrected ERP peak amplitudes (T0 minus baseline) with ccPAS (Exp5$_{STS-V1}$, Ctrl$_{V1-STS}$) as a between-subjects factor and Electrode cluster (left, right) as a within-subjects factor. Post-hoc analysis was performed using the Duncan test to correct for multiple comparisons. The Greenhouse–Geisser correction was employed where appropriate. In all the analyses, partial eta squared ($\eta_p^2$) was computed as a measure of effect size for the main effects and interactions, whereas *Cohens' d* was computed for *t*-tests and post-hoc comparisons. All statistical tests were two-tailed and conducted using Statistica v. 12 (StatSoft, Inc., Tulsa).

For source analysis, we estimated current source densities by projecting scalp potentials into source space using the sLORETA method[77,78], as implemented in Brainstorm software[97], and the ICBM 152 MRI template, as in Experiment 1. To investigate the effect of ccPAS at the source level, we obtained within-group source activation differences (T0 minus baseline) separately for each participant. For the statistical analysis, following the recommended procedures for unconstrained source analysis, we compared the power for each source and computed between-groups power tests (F-tests). We focused on the three time windows already analyzed at the sensor level for the P1, N170 and P2 components and performed three separate statistical tests, one for each time window. The source activation values at all timepoints within each time window were averaged before the analysis. To correct for multiple comparisons, we used the False Discovery Rate method (FDR)[98].

### Reporting summary

Further information on research design is available in the Nature Portfolio Reporting Summary linked to this article.

## Data availability

The experimental stimuli and all data analyzed in this study have been deposited on Open Science Framework (OSF) and can be accessed here: https://osf.io/yqbsj/. Source data are provided with this paper.

## Code availability

The custom-made MATLAB scripts used for data collection have been deposited on Open Science Framework (OSF) and can be accessed here: https://osf.io/yqbsj/.

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

## Acknowledgements

Work supported by #NEXTGENERATIONEU (NGEU) and funded by the Ministry of University and Research (MUR), National Recovery and Resilience Plan (NRRP), project MNESYS [PE0000006] —A Multiscale integrated approach to the study of the nervous system in health and disease (DN. 1553 11.10.2022) awarded to A.A. This work was also supported by grants from the FISM – Fondazione Italiana Sclerosi Multipla [2022/R-Single/071] financed or cofinanced with the '5 per mille' public funding, awarded to A.A; Bial Foundation [33/2022] awarded to S.B. and [304/2022] awarded to A.A.; Fondazione del Monte di Bologna e Ravenna [1402bis/2021], and Universidad Católica Del Maule [CDPDS2022] awarded to A.A.; European Research Council (ERC) Consolidator Grant 2017 "LIGHTUP" [772953] awarded to M.T. We would like to express our gratitude to Brianna Beck and Sonia Turrini for providing valuable feedback on the manuscript. Additionally, we thank Sonia Turrini's assistance in preparing the illustrations.

## Author contributions

A.A. conceptualized the study. S.B and A.A. designed the experiments. S.B., M.Z. and P.D.L. conducted the experiments. S.B., M.Z., P.D.L., A.C., G.A. and A.A. analyzed the data. S.B., G.A., V.R., M.T. and A.A. interpreted the findings. S.B., M.T. and A.A. wrote the manuscript. M.Z., G.A. and V.R. provided feedback and revised the manuscript. All the authors read and approved the final manuscript.

## Competing interests

The authors declare no competing interests.
