## [Peer Review File · Nature Communications]

Increasing associative plasticity in temporo-occipital back-projections improves visual perception of emotions from facial stimuliREVIEWER COMMENTS

Reviewer #1 (Remarks to the Author):

Overall this is a fascinating work. Borgomaneri and colleagues use a novel combination of network-based TMS causal approaches, i.e., TMS-EEG coregistration and ccPAS, to test the critical role and neuroplasticity of backward STS-V1 connectivity on the ability to perceive emotional expressions in noisy conditions. The authors first use TMS-EEG to establish the timing of STS-V1 interactions and then, building on the evidence of long-latency interactions, they design a novel information-based long-latency ccPAS protocol to drive Hebbian associative plasticity over STS-V1 projections. The noteworthy result is that administering ccPAS aimed at strengthening STS-to-V1 back-projections enhances participants' ability to discriminate facial expressions. This behavioral effect was replicated in two independent samples and experiments controlling for timing, order of TMS stimulation, and TMS unspecific factors. This allows authors to rule out alternative hypotheses and highlight the causal role of backward/recursive networks in accurate perception of emotional faces.

Moreover, while driving associative plasticity in the STS-V1 network impacts on recognition of facial expressions, it leaves gender recognition unaffected, as shown in a difficulty-matched control task, thus demonstrating functional specificity of the targeted network.

I enjoyed reading this well written manuscript. The conceptual framework and results are novel and very interesting. Previous studies in humans and monkeys have commonly attributed a prominent role to STS in face perception but did not formally test whether STS exerts its role via interactions with early visual cortices. The role of reentrant and recursive networks in perception and visual awareness has been hypothesized since the seminal work of Lamme and colleagues. However, to my knowledge, no prior study addressing the functions of the ventral visual system has performed direct causal testing of these hypotheses. Thus, a significant advance of this study is the use of powerful causal methods to demonstrate the critical role of STS-to-V1 back-projections in the perception of biologically relevant and complex stimuli, such as facial emotional expressions. The study provides unprecedented evidence that STS-V1 interactions are not epiphenomenal, but causally relevant to visual perception. Moreover, the study demonstrates that STS-V1 back-projections are functionally malleable.

The combination of TMS-EEG and ccPAS protocols is elegant and methodologically clever: I really like the idea of designing novel ccPAS protocols using an information-based TMS-EEG approach to induce spike-time-dependent-plasticity between targeted areas. This approach is tested against alternative hypotheses across 9 independent experimental groups. Generally speaking, the study is well designed and meets high standards. Analyses are thorough (but see a few minor points below) and conclusions are justified on the basis of a substantial amount of evidence.

I believe this work is of broad significance and interest to the fields of cognitive and behavioral neuroscience, neurophysiology and brain plasticity. It provides a novel example of a multi-site information-based TMS approach and would be relevant to theories of visual perception in general and models of facial expressions recognition in particular. In sum, I think this study could represent an important milestone for these fields of research.

This being said, I would ask the authors to clarify some conceptual and methodological aspects of their work and address some minor issues.

Here is a list of comments:

1) In comparison to many other TMS effects, the effects of this protocol are quite protracted, lasting at least 80 min. It might be worth pointing out that a similar time course of effects has been observed in physiological ccPAS experiments (e.g. Buch et 2011). It should be noted that in Exp2 and 3, statistical analyses suggest no consistent modulation of baseline-corrected d' between 0 and 80 ms post ccPAS. However, because LTP effects typically show inverted U-shaped temporal profiles, it would be informative to check if at least a trend can be detected here. Therefore, I would recommend to add a supplementary figure displaying the time course of the behavioral effect in the two experimental groups of Exp2 and 3, maybe using raw d' values.

2) Authors argue they have stimulated back-projections from pSTS to V1/V2. Although I agree this is likely the case, I feel authors should consider and discuss the possibility to have recruited wider networks converging into V1/V2. This does not take anything away from the novelty and interest of the results and would still be in line with the author's interpretation in terms of recursive processing. This is an impressive work with different methods, 9 experimental groups, and 119 participants and I want to clarify that I am not requesting additional evidence. However, I think authors should elaborate more on the possible recruitment of larger and indirect cortico-cortical pathways.

3) Feedback projections and recursive processing are thought to be fundamental for visual awareness (e.g., Pascual-Leone and Walsh, 2011 Science; Silvanto et al. 2005 Nat Neurosci) and yet, authors seem to interpret their findings in terms of visual perception, without addressing the concept of visual awareness more broadly. Please comment on this.

4) Prior evidence that rTMS over pSTS can hinder recognition accuracy of emotional faces (e.g., Pitcher et al. 2014 J Neurosci) should be more discussed in the text, also to emphasize the novelty of the present multi-site approach. In principle, the present TMS-EEG and ccPAS evidence can shed light on disruptive effects of STS-rTMS as such effects could be mediated by recursive STS-V1 networks.

5) Sample size of each experimental group looks in line with prior work and yet, authors do not clarify how the sample size was chosen. Also, did the authors remove any participants? Did they test the same subjects in more experiments? This should be reported.

6) Were participants randomly allocated to the different experimental groups? Did the authors use a double-blind approach, whereby participants and researchers who collected behavioral data blind to TMS conditions in Exp2-4?

7) Authors report effect sizes for the main ANOVAs, however, I would also recommend adding effect sizes for posthoc comparisons. In the figures, authors may want to use one or more asterisks to differentiate between comparisons meeting different statistical thresholds (e.g. $p < 0.05$, $p < 0.01$, or $p < 0.001$)

8) Lastly, I would recommend plotting individual data for Exp2-4, either in separate supplementary figures or by adjusting figures in the main text.

Reviewer #2 (Remarks to the Author):

The authors report a series of elegantly designed TMS experiments that investigate the connectivity between V1 and pSTS for expression recognition. I very much like this manuscript and support publication,. My main comment regards the timing of the information processing between the areas which I outline below.

1. The authors cite Pitcher 2014 in the methods regarding TMS site localisation but they do not engage with Experiment 2 in that paper which showed that double pulse delivered at 60-100ms and 100-140 ms after stimulus onset impaired task performance but TMS delivered at 130-170ms and 170-210ms did not impair task performance.

This suggests that visual information feed forward from early visual cortex to pSTS in less than 100ms and is not behaviourally relevant at 200ms.

So can the authors please explain why 200ms is an optimal duration for the feedback connections from pSTS to V1. That seems too long a latency to me. Because again information can go all over the brain in 200ms. Maybe this also relates to using ERP signals to establish temporal latencies for TMS experiments. Is it the case that ERP peaks and TMS stimulation latencies should directly correlate?

I am not sure I know the answer to this question but given prior studies have chronometric TMS to investigate when the pSTS processes facial expressions I think it is appropriate for the authors to address this work.

Minor comments

1. Opening sentence - While I can understand that to call V1/V2 a critical node for expression processing is technically correct. It is equally correct to say that V1/V2 is a critical node for perceiving any visual stimulus. Maybe consider reframing that sentence.

2. Regarding stimulation targeting of the pSTS site. The authors can cite Sliwinska and Pitcher 2018 which shows the established inter subject variability of the face-selective area in the pSTS and how their site is within that range.

REVIEWER COMMENTS

Reviewer #1 (Remarks to the Author):

Overall this is a fascinating work. Borgomaneri and colleagues use a novel combination of network-based TMS causal approaches, i.e., TMS-EEG coregistration and ccPAS, to test the critical role and neuroplasticity of backward STS-V1 connectivity on the ability to perceive emotional expressions in noisy conditions. The authors first use TMS-EEG to establish the timing of STS-V1 interactions and then, building on the evidence of long-latency interactions, they design a novel information-based long-latency ccPAS protocol to drive Hebbian associative plasticity over STS-V1 projections. The noteworthy result is that administering ccPAS aimed at strengthening STS-to-V1 back-projections enhances participants' ability to discriminate facial expressions. This behavioral effect was replicated in two independent samples and experiments controlling for timing, order of TMS stimulation, and TMS unspecific factors. This allows authors to rule out alternative hypotheses and highlight the causal role of backward/recursive networks in accurate perception of emotional faces.

Moreover, while driving associative plasticity in the STS-V1 network impacts on recognition of facial expressions, it leaves gender recognition unaffected, as shown in a difficulty-matched control task, thus demonstrating functional specificity of the targeted network. I enjoyed reading this well written manuscript. The conceptual framework and results are novel and very interesting. Previous studies in humans and monkeys have commonly attributed a prominent role to STS in face perception but did not formally test whether STS exerts its role via interactions with early visual cortices. The role of reentrant and recursive networks in perception and visual awareness has been hypothesized since the seminal work of Lamme and colleagues. However, to my knowledge, no prior study addressing the functions of the ventral visual system has performed direct causal testing of these hypotheses. Thus, a significant advance of this study is the use of powerful causal methods to demonstrate the critical role of STS-to-V1 back-projections in the perception of biologically relevant and complex stimuli, such as facial emotional expressions. The study provides unprecedented evidence that STS-V1 interactions are not epiphenomenal, but causally relevant to visual perception. Moreover, the study demonstrates that STS-V1 back-projections are functionally malleable.

The combination of TMS-EEG and ccPAS protocols is elegant and methodologically clever: I really like the idea of designing novel ccPAS protocols using an information-based TMS-EEG approach to induce spike-time-dependent-plasticity between targeted areas. This approach is tested against alternative hypotheses across 9 independent experimental groups. Generally speaking, the study is well designed and meets high standards. Analyses are thorough (but see a few minor points below) and conclusions are justified on the basis of a substantial amount of evidence.

I believe this work is of broad significance and interest to the fields of cognitive and behavioral neuroscience, neurophysiology and brain plasticity. It provides a novel example of a multi-site information-based TMS approach and would be relevant to theories of visual perception in general and models of facial expressions recognition in particular. In sum, I

think this study could represent an important milestone for these fields of research. This being said, I would ask the authors to clarify some conceptual and methodological aspects of their work and address some minor issues.

We thank Reviewer 1 for considering our work of broad significance and interest and for his/her cogent and constructive comments. In the present revision we have addressed all the points raised by the reviewer and we feel the manuscript is improved as a result. We have followed the recommendations of the reviewer and, importantly, we have added a novel study (Experiment 5 in the revised manuscript) that combines ccPAS with ERPs during the main behavioral task. The new experiment further strengthens the results of the study and the conclusions that can be drawn. Indeed, we further replicate the main behavioral findings already observed in Experiment 2 and 3, and show the neurophysiological correlates of improved perception of emotional expressions. The new experiment in particular demonstrates that our long-latency ccPAS protocol enhanced processing of emotional faces, particularly over early visual areas, i.e., the site of convergent activation during pSTS-to-V1/V2 ccPAS, consistent with an Hebbian STDP mechanism.

Reviewer #1, comment 1:

In comparison to many other TMS effects, the effects of this protocol are quite protracted, lasting at least 80 min. It might be worth pointing out that a similar time course of effects has been observed in physiological ccPAS experiments (e.g. Buch et 2011). It should be noted that in Exp2 and 3, statistical analyses suggest no consistent modulation of baseline-corrected d' between 0 and 80 ms post ccPAS. However, because LTP effects typically show inverted U-shaped temporal profiles, it would be informative to check if at least a trend can be detected here. Therefore, I would recommend to add a supplementary figure displaying the time course of the behavioral effect in the two experimental groups of Exp2 and 3, maybe using raw d' values.

Authors' reply to Reviewer #1, comment 1:

We thank Reviewer 1 for prompting us on this issue. It is indeed the case that LTP-like effects reported in prior ccPAS studies have often reported an inverted U-shaped temporal profile – or at least a trend – with little or no effects immediately following ccPAS (at T0), stronger effects between 30 and 60 min, and smaller or no effects after 90 minutes (e.g., Rizzo et al. 2009; Fiori et al. 2018; Chao et al. 2015; Romei et al. 2016). Nevertheless, there is variability across studies, with some reporting strong effects already at T0 and attenuated effects only following 3h from ccPAS (e.g., Buch et al. 2011), and others reporting short-lasting effects (e.g., Koch et al. 2013; Chiappini et al. 2020).

The reviewer is correct in noting that we observed an increase in performance following ccPAS, but no significant differences between time points in the explored post-ccPAS time window.

To further explore possible trends in the temporal unfolding of ccPAS effects, we followed the Reviewer's suggestion and pooled together individual's raw d' values from all participants undergoing the critical ccPAS manipulation – i.e., the pSTS-to-V1 ccPAS –

that now includes a further experiment (i.e., Experiment 2, 3 and 5).

Although we confirm no significant differences between post-ccPAS timepoints, curve inspection suggests a gradual numerical increase from T0 to T60 followed by a slight decline at T80, just as suggested by the Reviewer. Following the Reviewer's recommendation, we have included the figure below in the supplementary material to discuss the observed trend.

Figure S2. Temporal dynamics of the behavioral aftereffects of ccPAS across Experiments 2, 3 and 5.

We further examined the temporal unfolding of the ccPAS-induced increase in sensitivity to facial expressions in the 17-ms exposure condition by pooling together raw d' data from the three experimental groups (i.e., Exp2_{STS-V1}, Exp3_{STS-V1}, Exp5_{STS-V1}). Compared to baseline, a significant increment in d' values was present throughout the 5 post-ccPAS sessions (i.e., T0, T20, T40, T60, T80; all $t \geq 3.79$, all $p < 0.001$, all Cohen's $d \geq 0.73$). No statistical differences between post-ccPAS sessions were found (all $t \leq 1.68$, all $p \geq 0.10$). This suggests prolonged effects of the ccPAS protocol targeting pSTS-toV1/V2 feedback connections for at least 80 minutes. Additionally, mean d' values over the 6 time bins (Baseline, T0, T20, T40, T60, T80) were accurately fitted to a quadratic regression function, defined by the equation $y = a + bx + cx^2$ ($y = 1.13 + 0.563*x - 0.058*x^2$; SSE = 0.0046; $R^2_{adj} = 0.99$). The fitting revealed peak performance enhancement 40-60 minutes post-ccPAS and a decay trend afterward.

In the discussion section we also state:

“The enfolding and long-lasting behavioral effect of ccPAS (see Figure S2) appears in keeping with prior TMS studies on Hebbian plasticity in humans^{28,33–36}. In fact, visual sensitivity was increased immediately after the stimulation protocol and persisted for at least 80 minutes, following an inverse U-shaped curve peaking at around 40-60 minutes.”

Reviewer #1, comment 2:

Authors argue they have stimulated back-projections from pSTS to V1/V2. Although I agree this is likely the case, I feel authors should consider and discuss the possibility to have recruited wider networks converging into V1/V2. This does not take anything away from the novelty and interest of the results and would still be in line with the author's interpretation in terms of recursive processing. This is an impressive work with different methods, 9 experimental groups, and 119 participants and I want to clarify that I am not requesting additional evidence. However, I think authors should elaborate more on the possible recruitment of larger and indirect cortico-cortical pathways.

Authors' reply to Reviewer #1, comment 2:

We agree with Reviewer 1 that our stimulation protocol may have affected wider networks converging into V1/V2 due to the long stimulation interval we used during ccPAS (i.e., 200 ms). We also appreciate that the reviewer considers our study impressive and recommends discussing the possible involvement of larger and indirect pathways.

However, given the novelty of our approach and the relevance of the research question, we decided to address the issue of ccPAS-induced network changes more directly. Rather than speculating on this issue, we decided to carry out a further experiment to provide more direct data on how neural processing of emotional faces is influenced by ccPAS and test whether improved perception is associated with enhanced activity of the targeted temporo-occipital areas or larger brain networks.

Therefore, in the present revision, we present an entirely new study (Experiment 5) testing the neural underpinnings of ccPAS behavioral effects on the main emotion discrimination task with EEG event-related potentials (ERPs). This new experiment enabled us to directly assess changes in brain response to emotional faces following STS-to-V1 ccPAS.

First, the new study replicates the behavioral findings of improved emotion discrimination already observed in Experiment 2 and 4 (see Figure 6C below). Remarkably, Experiment 5 also shows that, following the critical ccPAS manipulation, the P1 component evoked by emotional faces during emotion discrimination is significantly enhanced (see Figure 6A and B). No similar changes, neither behavioral or physiological, were observed following a control ccPAS protocol (V1-to-STS ccPAS).

Figure 6. Physiological and behavioral changes following long-latency pSTS-to-V1/V2 ccPAS in Experiment 5. (A) Electrode clusters in the left unstimulated (PO7, P7, P5) and right stimulated hemispheres (PO8, P8, P6) showing grand average ERP waveforms pre-ccPAS (dashed line) and post-ccPAS (T0, continuous line); (B) P1 and d' mean values at T0 expressed relative to baseline values. Dots represent individual data. Red and black asterisks indicate significant increases relative to baseline and to other T0 conditions, respectively; * $p \leq 0.05$, ** $p \leq 0.01$; *** $p \leq 0.001$. Error bars denote SEM.

Source reconstruction of P1 enhancement following the experimental rather than the control ccPAS protocol highlighted the involvement of early occipito-temporal areas with maximal activations near the targeted V1/V2 site.

Figure 7. Localization of physiological changes following long-latency pSTS-to-V1/V2 ccPAS in Experiment 5. (A) Scalp map distribution showing changes in P1 amplitude in the Exp5_{ST5-V1} and Ctrl_{V1-ST5} groups following ccPAS (T0 vs. baseline). (B) Scalp distribution and source reconstruction showing higher increases in activity in the Exp5_{ST5-V1} group compared to the Ctrl_{V1-ST5} group (FDR-corrected $p \leq 0.005$). The increase in activity was maximal over early occipital regions, with the peak of activation at Talairach coordinates $x = 12$, $y = -98$, $z = -3$.

While these findings do not rule out that larger network may have been recruited during ccPAS, they show that significant neural aftereffects of ccPAS occurred maximally in V1/V2, the site of convergent activation during ccPAS protocol, consistent with the notion of Hebbian STDP.

We have discussed this topic in several passages of the revised discussion, including the following:

“The improved ability to discriminate emotional expressions was paralleled by an increase in the P1 component amplitude post-ccPAS, reflecting strengthened temporo-occipital activations during early stages of visual processing. This electrophysiological biomarker, though measurable in pSTS, was maximal over V1/V2 where neural activations were expected to converge due to ccPAS targeting of pSTS-to-V1/V2 back-projections. In keeping with this idea, the neuronal generators of the enhanced P1 activity nicely corresponded to the coordinates of the stimulated sites during ccPAS and were not found following stimulation of the same sites in forward direction (V1/V2-to-pSTS).”

(....)

We cannot rule out the possibility that the ccPAS protocol with a 200-ms ISI also indirectly recruited larger networks besides occipito-temporal areas. Indeed, TMS effects can be site- and function-specific, but not necessarily site-limited^{24,25}. Yet, we did not find support for this possibility, as the EEG results in Experiment 5 located the effects of ccPAS within the stimulated sites of the occipito-temporal network, primarily in V1/V2, as predicted by Hebbian STDP.”

Reviewer #1, comment 3:

Feedback projections and recursive processing are thought to be fundamental for visual awareness (e.g., Pascual-Leone and Walsh, 2011 *Science*; Silvanto et al. 2005 *Nat Neurosci*) and yet, authors seem to interpret their findings in terms of visual perception, without addressing the concept of visual awareness more broadly. Please comment on this.

Authors' reply to Reviewer #1, comment 3:

Thanks for prompting us on this point. We agree that backward projections are functionally relevant not only for perceptual decision making but also visual awareness. Indeed, the seminal work of Pascual-Leone and Walsh and Silvanto et al. have provided compelling evidence that interference with V1 activity disrupts awareness of TMS-induced motion phosphene following V5 stimulation at specific time points. These findings highlighted the dynamics and causal relevance of V5-to-V1 interactions in visual awareness. Yet, the same research group (Koivisto et al. 2010 *Neuroimage*) has more recently proposed that extrastriate-V1 feedback is not specific to visual awareness and may serve more general and basic visual processing. We have now referred to the literature outlining the functional role of feedback projections in visual awareness in the introduction and discussion, as recommended by this reviewer. However, since we did not directly test visual awareness in our participants and on a trial-by-trial basis, we would prefer remain cautious on this point and opt for a sober interpretation of our results as reflecting enhanced perception/recognition. Clearly, the relevance of our protocol to promote awareness, as

the Reviewer suggests, is paramount. We are actively pursuing this by measuring awareness changes directly and testing patients with V1 damage.

Reviewer #1, comment 4:

Prior evidence that rTMS over pSTS can hinder recognition accuracy of emotional faces (e.g., Pitcher et al. 2014 J Neurosci) should be more discussed in the text, also to emphasize the novelty of the present multi-site approach. In principle, the present TMS-EEG and ccPAS evidence can shed light on disruptive effects of STS-rTMS as such effects could be mediated by recursive STS-V1 networks.

Authors' reply to Reviewer #1, comment 4:

We thank Reviewer 1 for his/her comment. We have tried to better cast our current results in the light of prior TMS work on emotion perception. In doing so, we have also pinpointed the differences between prior approaches and the current one in the following passage of the discussion:

“Likewise, pSTS shows greater activation when participants are asked to discriminate facial expressions^{53,54} than when they are asked to discriminate faces based on morphological cues. Indeed, visual discrimination of facial identity, age, or gender relies more on ventral occipito-temporal face-selective areas⁴¹⁻⁴³. Accordingly, repetitive TMS studies have provided causal evidence for a dissociation between OFA and pSTS in discriminating faces based on morphological vs. expressive features, respectively⁵⁵⁻⁵⁹. Our study expands these prior findings by providing evidence of a functionally specific cortico-cortical neural mechanism through which pSTS can contribute to accurate recognition of emotional facial expressions through a top-down influence over early visual cortex.”

Moreover, to emphasize the differences between our approach and those used in prior rTMS studies (e.g. Pitcher et al. 2014) we have added the following passage:

“Rapid feedback interactions coexist with the initial and dominant feedforward signal flow and can influence basic levels of visual processing⁶⁰⁻⁶³. Likewise, early interference with pSTS through rTMS in the 60-140 ms time window impairs ongoing discrimination of facial expressions⁵⁵. These findings suggest that pSTS exerts its influence on emotion processing at multiple stages. This does not contradict our results. In fact, the early influence of pSTS on perception was reported during a single site, online rTMS interference protocol in an emotional face recognition task⁵⁵, whereas the long latency of our ccPAS protocol referred to the time window of maximal pSTS-to-V1/V2 interactions evidenced by single pulse TMS-EEG co-registration at rest (Experiment 1). The ccPAS protocol itself was administered offline in Experiments 2-5, and its behavioral impact was measured in a subsequent task. Further evidence of multiple temporal windows of pSTS influence also comes from Experiment 1, where we detected faster (although smaller and short-lasting) effects of pSTS stimulation on activity in early visual areas. These findings may inform further ccPAS manipulations based on rapid pSTS-V1/V2 feedback interactions, and future research could leverage both short- and long-latency neural interactions to promote Hebbian plasticity.”

Reviewer #1, comment 5:

Sample size of each experimental group looks in line with prior work and yet, authors do not clarify how the sample size was chosen. Also, did the authors remove any participants? Did they test the same subjects in more experiments? This should be reported.

Authors' reply to Reviewer #1, comment 5:

We have now added all the relevant information in the main text, and we thank the reviewer for helping us improve the standards of our report.

Regarding sample size estimation in Exp1, we could not perform a power analysis because of a lack of prior studies testing the influence of STS stimulation over V1/V2. Therefore, the sample size of 10 participants was determined based on previous TMS-EEG co-registration studies investigating the visual system (e.g., Zanon et al. 2009, N=9; Thut et al. 2011 Curr Biol, N=8; Romei et al. 2012 Curr Biol, N=9; Koivisto et al. 2017 Neuropsychologia, N=12; Zazio et al. 2019 Brain Top, N=8; see also the recent work of Veniero et al. 2021 Nat Comm, N=11). We have quoted relevant papers in the manuscript to support our choice.

There is no straightforward way to perform a power analysis for complex designs with more than two factors, and no prior ccPAS study tested pSTS-V1/V2 areas on emotion recognition. Therefore, we estimated the sample of the experimental groups in Exp2 and 3 based on prior work in our lab investigating the effect of V5-V1 ccPAS on motion perception (Romei et al. 2016 Curr Biol; Chiappini et al. 2018 Curr Biol) and the effect of STS-rTMS on emotion perception (Paracampo et al. 2018 Neuropsychologia). All these studies showed large effect sizes (mean Cohen's $d = 1.10$). Using G*Power 3 software (Faul et al. 2007) with power $(1-\beta) = 0.95$ and $\alpha = 0.05$, we estimated that a sample of 11 participants would be sufficient to show baseline vs. post-ccPAS differences in the experimental groups. We slightly increased this sample to 13/14 participants for each experimental or control group in Exp 2-4. Moreover, we increased the sample to 18 participants for each group of Experiment 5, testing behavioral and electrophysiological (EEG) data. The resulting sample sizes (N=42 in Exp2, N=32 in Exp3, N=28 in Exp4, N=36 in Exp5) were similar or larger than prior STS-rTMS studies on emotion perception (Pitcher et al. 2014 J Neurosci, N=10 and N=12; Candidi et al. 2015 Cortex, N=16; Paracampo et al. 2018 Neuropsychologia, N=16; Sliwinska & Pitcher 2018 Neuroimage, N=30; Ferrari et al. 2018 Cogn Aff Behav Neurosci, N=36; Pitcher et al. 2020 Cereb Cortex, N=14).

Four participants were excluded in the initial phases of the experiments 2-4 because of technical failures during ccPAS or before it. In Experiment 5, we had a technical failure in recording part of the behavioral response of one participant and have therefore removed behavioral data of this participant from the analysis.

Finally, we did not test the same participants in more than one experiment.

Reviewer #1, comment 6:

Were participants randomly allocated to the different experimental groups? Did the authors

use a double-blind approach, whereby participants and researchers who collected behavioral data blind to TMS conditions in Exp2-4?

Authors' reply to Reviewer #1, comment 6:

We thank Reviewer 1 for asking on this point. We have now added the following passage in the Methods section of the main text.

“For behavioral data, we implemented a double-blind procedure: participants were blinded to group allocation, and the experimenters who collected and analyzed the data were blinded to the ccPAS conditions. The experimenters who administered ccPAS were not blinded to group allocation because they had to set TMS parameters (i.e., order of pulses, ISI, and orientation of the coils).”

Reviewer #1, comment 7:

Authors report effect sizes for the main ANOVAs, however, I would also recommend adding effect sizes for posthoc comparisons. In the figures, authors may want to use one or more asterisks to differentiate between comparisons meeting different statistical thresholds (e.g. $p < 0.05$, $p < 0.01$, or $p < 0.001$)

Authors' reply to Reviewer #1, comment 7:

We thank Reviewer 1 for his/her suggestion. We have now computed Cohen's d values for post-hoc comparisons (Cohen, 1977; Wolf, 1986).

Moreover, we have now modified the number of asterisks in all our figures to differentiate between comparisons meeting different statistical thresholds, as recommended.

See for example the revised Figure 3 below:

Figure 3. Experiment 2 results showing a selective increase in visual sensitivity (d') to emotional facial expressions when faces are exposed for 17 ms. The increase is specific to the ccPAS protocol that targets long-latency pSTS-to-V1/V2 backward connections using the critical ISI of 200 ms (i.e., Exp2_{STS-V1}). No change is observed following ccPAS protocols controlling for the timing of the paired stimulation with ISIs of 0 ms (Ctrl_{0ms}) or 100 ms (Ctrl_{100ms}). Mean d' prime values across the post-ccPAS time points (average of T0, T20, T40, T60, T80) are expressed relative to baseline values. Dots represent individual data. Red asterisks indicate a significant increase in the post-ccPAS condition relative to baseline; black asterisks denote significant differences relative to other post-ccPAS conditions; * $p \leq 0.05$, ** $p \leq 0.01$, *** $p \leq 0.001$. Error bars denote SEM.

Reviewer #1, comment 8:

Lastly, I would recommend plotting individual data for Exp2-4, either in separate supplementary figures or by adjusting figures in the main text.

Authors' reply to Reviewer #1, comment 7:

We thank Reviewer #1 for her/his suggestion. We have now adjusted the figures in the main text, plotting the individual data as shown above.

Reviewer #2 (Remarks to the Author):

The authors report a series of elegantly designed TMS experiments that investigate the connectivity between V1 and pSTS for expression recognition. I very much like this manuscript and support publication,. My main comment regards the timing of the information processing between the areas which I outline below.

We thank the reviewer for considering our study elegantly designed and supporting its publication. In the present revision, we have addressed all the points raised by the reviewer, and we feel the manuscript is significantly improved as a result.

Reviewer #2, comment 1:

The authors cite Pitcher 2014 in the methods regarding TMS site localisation but they do not engage with Experiment 2 in that paper which showed that double pulse delivered at 60-100ms and 100-140 ms after stimulus onset impaired task performance but TMS delivered at 130-170ms and 170-210ms did not impair task performance. This suggests that visual information feed forward from early visual cortex to pSTS in less than 100ms and is not behaviourally relevant at 200ms.

So can the authors please explain why 200ms is an optimal duration for the feedback connections from pSTS to V1. That seems too long a latency to me. Because again information can go all over the brain in 200ms. Maybe this also relates to using ERP signals to establish temporal latencies for TMS experiments. Is it the case that ERP peaks and TMS stimulation latencies should directly correlate?

I am not sure I know the answer to this question but given prior studies have chronometric TMS to investigate when the pSTS processes facial expressions I think it is appropriate for the authors to address this work.

Authors' reply to Reviewer #2, comment 1:

We thank Reviewer 2 for prompting us on this point. As also recommended by Reviewer 1, we have now discussed the findings of Experiment 2 reported by Pitcher et al., 2014, showing early disruption of face processing by STS interference. This also allowed us to discuss the different methodological approaches and general framework used here compared to those of Picher et al., 2014, which prevent straightforward comparisons and may account for the different findings.

Pitcher et al., 2014 tested the effect of online repetitive TMS during the perception of emotional facial expressions while here we are recursively stimulating (employing a different dual-site TMS protocol) offline interactions between two areas that may take place at different timings. Indeed, we agree with the reviewer that feedback interactions can occur at shorter latencies also. Please note that with our TMS-EEG protocol, we captured even faster STS-V1 interactions. However, the apex of the effect was observed at 200 ms. Thus, we used such a time window to create the optimal condition for our protocol.

We have discussed the issue of temporal dynamics in a few passages of the discussion, including the following:

“Evidence from non-human primates shows that the dominant direction of signal flow within occipito-temporal networks is feedforward shortly after stimulus onset (i.e., within 150-200 ms), but then gradually reverses to feedback at longer latencies and remains so at rest^{20,60}. The long-latency 200-ms ISI of the critical ccPAS protocol is in keeping with these neural dynamics and suggests that its efficacy depends upon the relative delay of backward communication within temporo-occipital networks^{60,61}. We cannot rule out the possibility that the ccPAS protocol with a 200-ms ISI also indirectly recruited larger networks besides occipito-temporal areas. Indeed, TMS effects can be site- and function-specific, but not necessarily site-limited^{24,25}. Yet, we did not find support for this possibility, as the EEG results in Experiment 5 located the effects of ccPAS within the stimulated sites of the occipito-temporal network, primarily in V1/V2, as predicted by Hebbian STDP.

Rapid feedback interactions coexist with the initial and dominant feedforward signal flow and can influence basic levels of visual processing^{60–63}. Likewise, early interference with pSTS through rTMS in the 60-140 ms time window impairs ongoing discrimination of facial expressions⁵⁵. These findings suggest that pSTS exerts its influence on emotion processing at multiple stages. This does not contradict our results. In fact, the early influence of pSTS on perception was reported during a single site, online rTMS interference protocol in an emotional face recognition task⁵⁵, whereas the long latency of our ccPAS protocol referred to the time window of maximal pSTS-to-V1/V2 interactions evidenced by single pulse TMS-EEG co-registration at rest (Experiment 1). The ccPAS protocol itself was administered offline in Experiments 2-5, and its behavioral impact was measured in a subsequent task. Further evidence of multiple temporal windows of pSTS influence also comes from Experiment 1, where we detected faster (although smaller and short-lasting) effects of pSTS stimulation on activity in early visual areas. These findings may inform further ccPAS manipulations based on rapid pSTS-V1/V2 feedback interactions, and future research could leverage both short- and long-latency neural

interactions to promote Hebbian plasticity.”

Reviewer #2, comment 2:

Minor comments, Opening sentence - While I can understand that to call V1/V2 a critical node for expression processing is technically correct. It is equally correct to say that V1/V2 is a critical node for perceiving any visual stimulus. Maybe consider reframing that sentence.

Authors' reply to Reviewer #2, comment 2:

We agree with Reviewer 2 that V1/V2 is a critical node for perceiving any visual stimulus rather than a critical node for processing emotional expressions. We have now rephrased the sentence as: “

“The posterior superior temporal sulcus (pSTS) is a critical node of a network specialized for perceiving emotional facial expressions that is reciprocally connected with early visual cortices (V1/V2).”

Reviewer #2, comment 3:

Regarding stimulation targeting of the pSTS site. The authors can cite Sliwinska and Pitcher 2018 which shows the established inter subject variability of the face-selective area in the pSTS and how their site is within that range.

Authors' reply to Reviewer #2, comment 3:

We thank Reviewer 2 for his/her helpful suggestion. We have now cited the work of Sliwinska and Pitcher (2018) as recommended. We rephrased the passage as follows: “The pSTS site is well in keeping with other brain imaging meta-analyses on emotional face perception^{53,89} and prior TMS studies^{56–58}, falling within the range of interindividual variability of the face-selective area in the pSTS reported by Sliwinska and Pitcher⁵⁹.”

Indeed, we can safely claim that our pSTS site is within the range of coordinates shown by Sliwinska and Pitcher. As reported in the manuscript, we selected the right pSTS site based on the ALE meta-analysis of Dricu & Frühholz (2016). Based on this meta-analytic work, we computed mean subject-weighted MNI coordinates for emotion evaluation (75 experiments, 1742 participants) and passive observation (20 experiments, 411 participants) of emotional facial expressions. We obtained the following coordinates: $x = 58.57$ (range: 51, 62), $y = -48.85$ (range: -35, -53), $z = 10.85$ (range: 0, 15). Then we converted the mean coordinates in Talairach ($x = 53$, $y = -49$, $z = 10$) and localized the corresponding site on the participants' scalp with our neuronavigation system.

As we previously reported, our pSTS coordinates appear slightly anterior (21mm) relative to the mean MNI coordinates reported by Pitcher et al. (2014), $x=52$, $y=-70$, $z=2$. On the other hand, our coordinates appear more consistent with the MNI coordinates reported by

Sliwinska and Pitcher: $x=55$ (range 43, 68), $y=-39.3$ (range -24, -52), $z=9$ (range -3, 33). All our coordinates deviate less than 1.5 standard deviations from Sliwinska and Pitcher's mean coordinates. Moreover, our coordinates entirely fall within the reported range of interindividual variability.

As we briefly mention in the revised manuscript, our coordinates appear in keeping with other brain stimulation studies targeting pSTS during perception of emotional expressions (Candidi et al. 2015; Paracampo et al. 2018; Ferrari et al. 2018) and prior meta-analysis investigating the neural correlates of emotional face perception (Fusar-Poli et al. 2009; Sabatinelli et al. 2011), as it can be seen below.

PRESENT STUDY	Talairach: $x=53, y=-49, z=10$	MNI: $x=59, y=-49, z=11$
Candidi et al. 2015,	Talairach: $x=52, y=-48, z=8$	
Paracampo et al. 2018,	Talairach: $x=48, y=-49, z=4$	
Ferrari et al. 2018,	Talairach: $x=52, y=-48, z=8$	
Fusar-Poli et al. 2009	Talairach $x=56, y=-44, z=4$	
Sabatinelli et al. 2011		MNI: $x=53, y=-50, z=4$

REVIEWERS' COMMENTS

Reviewer #1 (Remarks to the Author):

I acknowledge the considerable work the authors did in revising their manuscript. They added several sections to the manuscript, added a new study, ran additional analyses, and solved all of the previous issues raised by reviewer 2 and me. I believe that the manuscript has been significantly improved as a result. The authors have followed all my recommendations and, notably, have included a new study (Experiment 5) that combines ccPAS and ERPs during the primary behavioral task. This addition reinforces the findings and conclusions of the study. In particular, the authors have replicated the main behavioral results from Experiments 2 and 3 and have provided neurophysiological evidence of the improved perception of emotional expressions. The new experiment demonstrates that the long-latency ccPAS protocol enhances the processing of emotional faces, particularly in early visual areas, which is consistent with the Hebbian STDP mechanism. The authors also answered Reviewer 2' insightful observations. In my opinion, the present revision addresses all of the previous issues raised and it is ready for acceptance.

Reviewer #2 (Remarks to the Author):

The authors have done a commendable job in responding to the comments from myself and the other reviewer. I support publication of the manuscript.